# MuSc: Zero-Shot Industrial Anomaly Classification and Segmentation with Mutual Scoring of the Unlabeled Images

**Xurui Li**[1,*], **Ziming Huang**[1,*], **Feng Xue**[3], **Yu Zhou**[1,2,†]
[1] School of Electronic Information and Communications, Huazhong University of Science and Technology
[2] Artificial Intelligence Research Institute, Wuhan JingCe Electronic Group Co.,LTD
[3] Department of Information Engineering and Computer Science, University of Trento
{xrli_plus,zmhuang,yuzhou}@hust.edu.cn, feng.xue@unitn.it

## Abstract

This paper studies zero-shot anomaly classification (AC) and segmentation (AS) in industrial vision. We reveal that the abundant normal and abnormal cues implicit in unlabeled test images can be exploited for anomaly determination, which is ignored by prior methods. Our key observation is that for the industrial product images, the normal image patches could find a relatively large number of similar patches in other unlabeled images, while the abnormal ones only have a few similar patches. We leverage such a discriminative characteristic to design a novel zero-shot AC/AS method by Mutual Scoring (MuSc) of the unlabeled images, which does not need any training or prompts. Specifically, we perform Local Neighborhood Aggregation with Multiple Degrees (LNAMD) to obtain the patch features that are capable of representing anomalies in varying sizes. Then we propose the Mutual Scoring Mechanism (MSM) to leverage the unlabeled test images to assign the anomaly score to each other. Furthermore, we present an optimization approach named Re-scoring with Constrained Image-level Neighborhood (RsCIN) for image-level anomaly classification to suppress the false positives caused by noises in normal images. The superior performance on the challenging MVTec AD and VisA datasets demonstrates the effectiveness of our approach. Compared with the state-of-the-art zero-shot approaches, MuSc achieves a **21.1%** PRO absolute gain (from 72.7% to 93.8%) on MVTec AD, a **19.4%** pixel-AP gain and a **14.7%** pixel-AUROC gain on VisA. In addition, our zero-shot approach outperforms most of the few-shot approaches and is comparable to some one-class methods. Code is available at https://github.com/xrli-U/MuSc.

## 1 Introduction

Industrial anomaly classification (AC) and anomaly segmentation (AS) are fundamental tasks in computer vision with diverse applications. Real-world anomalies may appear on a wide range of objects and textures. The high diversity of anomalies makes the AC/AS task challenging.

Many excellent unsupervised AC/AS approaches, e.g., (Zavrtanik et al., 2022; Zhang et al., 2023; Zavrtanik et al., 2021; Liu et al., 2023), have been proposed. They typically rely on the whole normal dataset for training, thus called as **full-shot** methods. In contrast, RegAD (Huang et al., 2022) and GraphCore (Xie et al., 2023) leverage a few normal images only, known as **few-shot** AC/AS setting, and also achieve promising accuracy. All the methods above can be categorized as one-class AC/AS approaches, and share a similar pipeline as shown in Fig. 1(a). Recently, WinCLIP (Jeong et al., 2023) and APRIL-GAN (Chen et al., 2023) propose **zero-shot** AC/AS, which breaks new ground by using text prompts for anomaly measurement, as shown in Fig. 1(b).

In these existing approaches, the test images are compared to the additionally introduced information, such as normal training images or text prompts. However, a rich amount of normal information

---

* Contributed Equally, † Corresponding Authors.

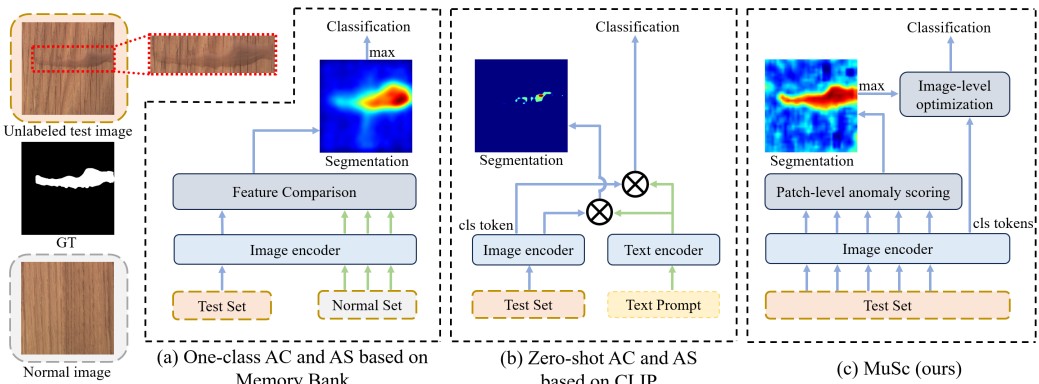

Figure 1: (a) One-class AC and AS based on a memory bank, which requires many normal reference images. (b) Zero-shot AC and AS based on CLIP, which relies on additional text prompts. (c) Our MuSc only leverages the unlabeled test images from patch-level and image-level.

implicit in the unlabeled test images is rarely used. (*According to statistics, the normal pixels accounted for* $97.26\%$ *in the test images of MVTec AD dataset and* $99.45\%$ *in VisA dataset*). Besides, due to the randomness and unpredictability of the anomalies, the abnormal regions may be dissimilar from each other, even though they belong to the same anomaly type. Therefore, both normal and abnormal cues in unlabeled test images can be exploited for anomaly determination.

Based on these observations, **we reveal that the anomaly can be detected by comparing it with other unlabeled test images, and hence we design a novel zero-shot AC/AS approach, named MuSc, which does not need any normal training images or prompts**. As shown in Fig. 1 (c), our approach consists of a patch-level representation extractor, a patch-level anomaly scoring approach, and an image-level optimization. First, we perform Local Neighborhood Aggregation with Multiple Degrees (LNAMD) on the patch token to express each image patch, which is capable of modeling anomalies in varying sizes and is suitable for AC and AS tasks. Next, we introduce a Mutual Scoring Mechanism (MSM), which leverages the unlabeled test images to assign the anomaly scores to each other. Such a simple mechanism generates a high-quality anomaly score for each image region. It can be credited to the observation that the normal image patches could find a large number of similar patches in other unlabeled images while the abnormal ones only have a few similar patches. Furthermore, we explore the relationship between image-level features and propose the Re-scoring with Constrained Image-level Neighborhood (RsCIN) to optimize the anomaly classification. Extensive experimental evaluations on MVTec AD and VisA datasets demonstrate the effectiveness of our approach. Compared with the state-of-the-art zero-shot AC/AS approaches, we achieve **+6.0%** AUROC classification metrics on MVTec AD and **+14.7%** increase on VisA. For anomaly segmentation, we obtain **+21.1%** PRO and **+21.9%** AP gains on MVTec AD and **+19.4%** AP gains on VisA. We further emphasize that our zero-shot approach outperforms most existing few-shot methods in 4-shot case and is comparable to several full-shot methods.

In summary, the major contributions of our work are:

- To the best of our knowledge, we propose the first method that only uses the unlabeled test images for industrial anomaly classification and segmentation.
- We reveal the potential capability of normal and abnormal patches contained in unlabeled images. It inspires us to propose a brand new AC/AS method based on mutual scoring of unlabeled images, which is a new mechanism for the community.
- Our approach significantly surpasses the existing zero-shot AC and AS approaches. In addition, our approach outperforms most of the few-shot approaches and is comparable to some full-shot methods.

## 2 RELATED WORKS

**Vision transformer.** Vision transformer (ViT) (Dosovitskiy et al., 2020) is widely used in multi-modal models and large vision models such as CLIP (Radford et al., 2021) and DINO (Caron et al.,

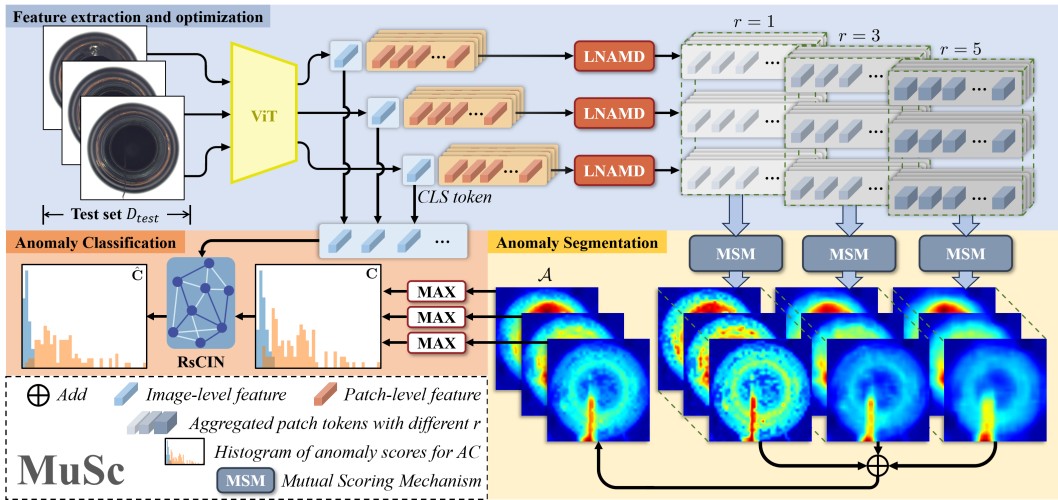

Figure 2: Our MuSc architecture. It consists of three parts: feature extraction and optimization (Section 3.1), MSM to obtain anomaly segmentation results (Section 3.2), and RsCIN to optimize classification results (Section 3.3).

2021). These models produce high-quality patch-level and image-level features. We find that the patch-level features have the limitation of detecting industrial anomalies of varying sizes. Swin transformer (Liu et al., 2021) proposes the varied-size window attention to calculate attention inside local windows of varying sizes, which can detect diverse size objects. We propose a simple method LNAMD to optimize the patch-level features so that it can detect varying size anomalies.

**Industrial anomaly classification and segmentation.** In the industrial vision field, previous works explore one-class AC/AS, such as PatchCore (Roth et al., 2022), PaDiM (Defard et al., 2021) and AST (Rudolph et al., 2023), which require a lot of normal reference images. Recently, some zero-shot methods have worked well. Most of them rely on text prompts for the image-text alignment such as WinCLIP (Jeong et al., 2023), Variance Vigilance Vanguard (Baugh et al., 2023) and SAA (Cao et al., 2023). In the few-shot setting, RegAD (Huang et al., 2022) and GraphCore (Xie et al., 2023) focus on data augmentation to estimate the normal feature distribution by a few reference normal images. These methods ignore the rich information in the unlabeled images. In the medical image analysis field, some methods like DDAD (Cai et al., 2023) and SRR (Yoon et al., 2021) integrate the unlabeled images into the normal image set for model training, which also needs to select a set of normal images. In the zero-shot AC/AS field, the method (Aota et al., 2023) explores the relationship between the patches inside one test texture image. ACR (Li et al., 2023) proposes a new adaptation strategy without human involvement and outperforms other methods. We propose the first zero-shot AC/AS method that only uses the unlabeled images.

**Manifold learning.** In high-dimensional manifolds, only local space maintains the nature of Euclidean space, so it is inaccurate to calculate the distance directly in high-dimensional manifolds. The idea of manifold learning methods (Zhou et al., 2003; Wang & Tu, 2012) is to construct a new embedding space where the distance metric matches the manifold structure. We find that the image-level features satisfy the conditions for higher dimensional manifolds, and hence we design the RsCIN based on manifold learning to optimize the pixel-level anomaly classification results.

## 3 METHOD

As illustrated in Fig. 2, our approach is directly designed on the unlabeled test images. Firstly, we extract the ViT feature for each unlabeled image and perform Local Neighborhood Aggregation with Multiple Degrees (LNAMD) on patch tokens to represent abnormal regions with varying sizes, detailed in Section 3.1. Secondly, we propose a Mutual Scoring Mechanism (MSM) in Section 3.2, in which the unlabeled images assign the anomaly scores to each other, resulting in pixel-level anomaly classification and segmentation. Thirdly, we leverage Re-scoring with Constrained Image-level Neighborhood (RsCIN) to optimize the pixel-level anomaly classification in Section 3.3.

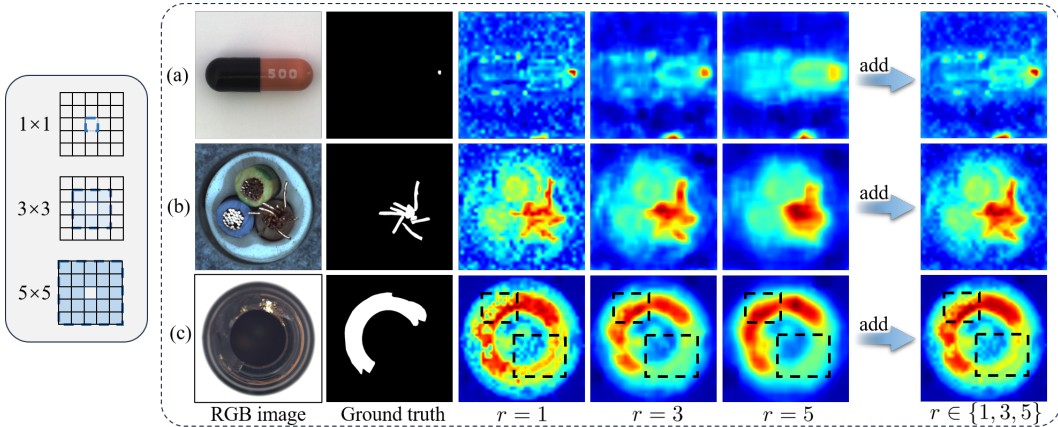

Figure 3: The visualization of anomaly segmentation results with different aggregation degrees $r$.

### 3.1 LOCAL NEIGHBORHOOD AGGREGATION WITH MULTIPLE DEGREES

Given the unlabeled test images $D_u = \{I_i, i = 1, ..., N\}$. We extract the ViT feature (Dosovitskiy et al., 2020) of $I_i$. To increase the abnormal modeling ability with varying sizes, we perform local neighborhood aggregation with multiple degrees on the patch tokens of ViT. Specifically, we define the patch tokens as $F_i \in \mathbb{R}^{M \times C}$ of $I_i$, where $M$ is the patch number, then we reshape $F_i$ to $\sqrt{M} \times \sqrt{M} \times C$. For each patch token, we follow (Roth et al., 2022) to use an adaptive average pooling operation in its $r \times r$ neighborhood to obtain the aggregated patch token $\hat{F}_i^r \in \mathbb{R}^{\sqrt{M} \times \sqrt{M} \times C}$ with different aggregation degree $r \in \{1, 3, 5\}$, shown in the left of Fig. 3, and we reshape it back to $M \times C$. Therefore each aggregated patch token is denoted as $\hat{p}_{i,l}^{m,r} \in \mathbb{R}^{1 \times C}$, where $m \in [1, M]$, we divide the layers of ViT into $L$ stages and $l \in \{1, 2, ..., L\}$ indicates the stage $l$ of ViT. We will integrate multiple aggregation degrees and stages of ViT in the next subsection.

Compared with using one patch size, the patch token with multiple aggregation degrees is suitable for representing anomalies with varying sizes, leading to high-quality anomaly scores even with a simple distance measurement. As the industrial images illustrated in Fig. 3. When $r = 1$, we do not perform LNAMD. The defect has a higher anomaly score in (a), which indicates that smaller $r$ is suitable for detecting small anomalies, while in (c), the large defect has several local false positives and false negatives, indicated by the black dotted box, since the smaller regions cannot fully express the entire defect. When $r = 5$, the false positives and false negatives are solved in (c), demonstrating that the aggregated feature is suited to detect the larger defects, while in (a), the small defect is smoothed and has a lower score. Therefore, using all aggregated patch tokens with different degrees achieves a good balance in detecting the anomaly with a wide range of sizes, which is the common case in real industrial scenes, as shown in the right side of Fig. 3.

### 3.2 MUTUAL SCORING MECHANISM OF THE UNLABELED IMAGES

In this subsection, we propose the mutual scoring mechanism to obtain the high-quality patch-level anomaly score by only using the unlabeled images in $D_u$, which consists of three steps. In the first step, we leverage each image in $\{D_u \backslash I_i\}$ to assign an anomaly score for each aggregated patch token of a test image $I_i$ in stage $l$ as follows,

$$a_{i,l}^{m,r}(I_j) = \min_n \|\hat{p}_{i,l}^{m,r} - \hat{p}_{j,l}^{n,r}\|_2 \tag{1}$$

where $r$ is the aggregation degree and $(I_j)$ indicates that image $I_j \in \{D_u \backslash I_i\}$ is employed for scoring. If the patch token $\hat{p}_{i,l}^{m,r}$ of $I_i$ is similar to any patch token $\hat{p}_{j,l}^{n,r}$ of $I_j$, the image $I_j$ assigns a small anomaly score to $\hat{p}_{i,l}^{m,r}$. Then we define the scoring vector $A_{i,l}^{m,r} = [a_{i,l}^{m,r}(I_1), a_{i,l}^{m,r}(I_2), ..., a_{i,l}^{m,r}(I_{N-1})]$ of each patch for each $l$ and $r$ according to $\{D_u \backslash I_i\}$. The statistical histogram of $A_{i,l}^{m,r}$ of all the normal patches in $D_u$ is given in Fig. 4(a), which indicates that most of the normal patches could find some similar patches in $D_u$. While (b) gives the histogram

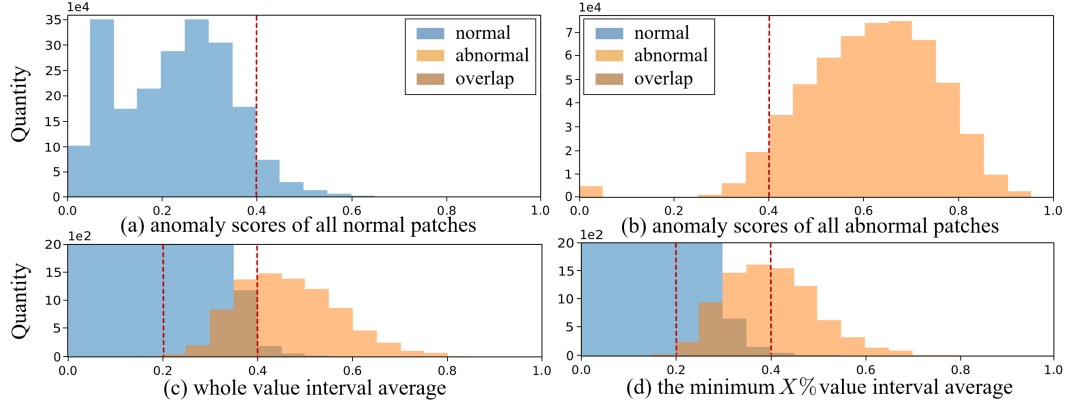

Figure 4: **Top**: The histograms of $A_{i,l}^{m,r}$ of all normal patches (a) and abnormal patches (b). **Bottom**: The histograms of $\bar{a}_{i,l}^{m,r}$ of normal patches and abnormal patches with the whole value interval (c) and the minimum $X\%$ value interval (d) to average. The **blue** and **orange** parts indicate the anomaly scores of normal patches and abnormal patches respectively under $r = 3$ and $l = 2$ setting.

of abnormal patches, which shows that most abnormal patches only have a few similar patches in $D_u$, indicating in the left corner of Fig. 4 (b), and they are dissimilar to most of the normal and other abnormal patches in (b).

Based on the above analysis, we claim that most of the unlabeled images in $D_u$ could assign a higher score for the anomaly patch and a lower score for the normal ones. Hence, directly performing an average operation on $A_{i,l}^{m,r}$ is capable of distinguishing the abnormal patches from the normal patches, shown in Fig. 4(c). For the overlap in (c), we notice that it is caused by some normal patches with appearance variations of different normal images, which are assigned higher scores by the dissimilar patches. To reduce these dissimilar patches in scoring, the second step of MSM is to perform an Interval Average (IA) operation on the minimum $X\%$ of $A_{i,m}^{l,r}$.

$$\bar{a}_{i,l}^{m,r} = \frac{1}{K} \sum_{k \in [1,K]} a_{i,l}^{m,r}(\bar{I}_k) \qquad (2)$$

where $\bar{I}$ is the images in the minimum $X\%$ value interval and $K$ is the number of these images. Such a design uses a small number of images to score each image patch, and suppresses the afore-mentioned issue effectively. As shown in Fig. 4(d), the anomaly score distribution of normal and abnormal patches has a smaller overlap in the score interval of [0.2, 0.4] compared with (c).

Aiming to achieve the fine-grained anomaly score for each image patch, the third step of MSM performs Eq. 1 and Eq. 2 on each combination of stage $l$ and aggregation degree $r$, respectively, and compute the average of all $\bar{a}_{i,l}^{m,r}$ to achieve the patch-level anomaly score as follows,

$$\mathbf{a}_i^m = \frac{1}{L} \sum_{l \in \{1,...,L\}} \frac{1}{3} \sum_{r \in \{1,3,5\}} \bar{a}_{i,l}^{m,r} \qquad (3)$$

The patch-level anomaly score vector of $I_i$ is defined as $\mathcal{A}_i = [\mathbf{a}_i^1, ..., \mathbf{a}_i^M]^\top$. Then we reshape $\mathcal{A}_i \in \mathbb{R}^{M \times 1}$ to $\sqrt{M} \times \sqrt{M} \times 1$ and upsample it to the original resolution of the input image as the anomaly segmentation result. Following existing approaches (Roth et al., 2022), we extract the maximum score $c_i = \max(\mathbf{a}_i^1, ..., \mathbf{a}_i^M)$ of image $I_i$ as its pixel-level anomaly classification score and the AC score vector of the images in $D_u$ is denoted as $\mathbf{C} = [c_1, ..., c_N]^\top$. See Appendix A.1.1 for more details.

## 3.3 CLASSIFICATION RE-SCORING WITH CONSTRAINED IMAGE-LEVEL NEIGHBORHOOD

We observe that the pixel-level AC score in $\mathbf{C}$ is sensitive to the local noises, e.g., in Fig. 5(i), a normal image that contains several small noises has a higher AC score $c = 0.129$, while the pure normal image (ii) has a lower score $c = 0.079$, when we calculate the class token distance between images (i) and (ii), they are almost the same with a similarity equal to 0.987, which reveals that the

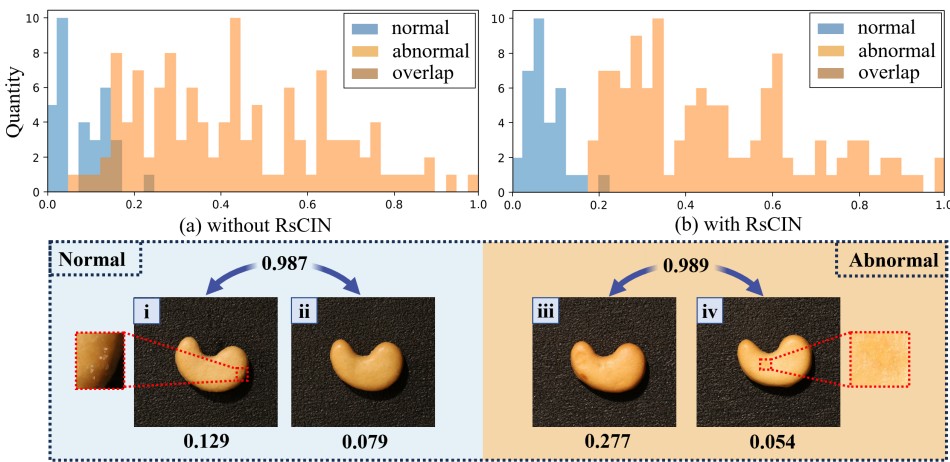

Figure 5: **Top**: Histogram of anomaly classification scores of unlabeled test images before (a) and after (b) using RsCIN, with **blue** representing normal images and **orange** representing abnormal images. **Bottom**: A normal and an abnormal example of RsCIN.

class token distance is less sensitive to local noises, and two images that have a higher class token similarity should have an approximate AC score. A similar situation can be observed in an abnormal image (iv), where the image contains a low contrast abnormal area and has a lower AC score in (iv), while it has a higher similar to the abnormal image (iii).

Motivated by such an observation, we propose the re-scoring with constrained image-level neighborhood to optimize the anomaly classification by exploiting the relationship between the image-level features of images in $D_u$. Specifically, given the class token $\mathcal{F}_i \in \mathbb{R}^{1 \times \mathcal{C}}$ of the last layer of ViT, we define an edge-weighted graph $G = (V, W)$, the vertexes $V$ denotes the images in $D_u$, and the edge weight $W$ is defined as the similarity of the class tokens $\mathcal{F}_i$ of $I_i$, i.e., $W_{i,j} = \mathcal{F}_i \cdot \mathcal{F}_j$, where $\cdot$ means dot product operation. Therefore, the manifold learning approaches, e.g., SSO (Jiang et al., 2011), can be employed for optimizing $\mathbf{C}$ by utilizing the token similarity $W$.

However, due to the high smoothness of $\mathbf{C}$, it may be influenced by a large number of images, as the experimental illustration in Sec. 4.2, excessive utilization of the images in $D_u$ may reduce the AC accuracy. Therefore, we propose a Multi-window Mask Operation (MMO) to constrain the image number, which makes each image only influenced by a small number of neighborhood images. To be specific, we design a binary window mask matrix $M_k \in \mathbb{R}^{N \times N}$ as,

$$M_k(i, j) = \begin{cases} 1, & \text{if } I_j \in \mathcal{N}_k(I_i) \\ 0, & \text{otherwise,} \end{cases} \tag{4}$$

$\mathcal{N}_k$ indicates the $k$-nearest neighbor. We use multiple $k$ to form the multi-window mask set $\overline{M} = \{M_{k_1}, .., M_{k_K}\}$, where $K$ is the number of window masks. Hence the AC score $\mathbf{C}$ is updated as,

$$\hat{\mathbf{C}} = (\sum_{M_k \in \overline{M}} (D^{-1}(M_k \odot W)\mathbf{C}) + \mathbf{C})/(K+1) \tag{5}$$

where $\hat{\mathbf{C}} \in \mathbb{R}^{N \times 1}$ is the optimized AC score vector, $D$ is a diagonal matrix with $D(i, i) = \sum_{j=1}^{N} M_k \odot W(i, j)$ and $\odot$ means element-wise multiplication. Eq. 5 utilizes the weighted AC scores of multiple neighbor images of $I_i$ to refine $\mathbf{C}$. Therefore $I_i$ is normal only if itself and its $k$ nearest neighbor images all have a small AC score. As shown in Fig. 5, the AC scores of normal and abnormal images in $\mathbf{C}$ have a large overlap in (a), while in (b), the overlap is reduced in $\hat{\mathbf{C}}$. See Appendix A.1.2 for more details. Meanwhile, we experimentally prove that the proposed image-level AC optimization method can be further employed to improve the AC accuracy of existing approaches in Appendix A.2.4.

## 4 EXPERIMENTS

**Implementation Details.** In this paper, we use ViT-L/14-336 pre-trained by OpenAI (Radford et al., 2021) as our backbone. It contains 24 layers, which are divided into 4 stages. Each stage is

Table 1: Quantitative comparisons on the **MVTec AD** and **VisA** datasets. We compare our MuSc with some state-of-the-art zero-shot and few-shot methods. Bold indicates the best performance, while underlined denotes the second-best result. All metrics are in %.

| Dataset | Method | Setting | AUROC-cls | F1-max-cls | AP-cls | AUROC-segm | F1-max-segm | AP-segm | PRO-segm |
|---|---|---|---|---|---|---|---|---|---|
| MVTec AD | WinCLIP (Jeong et al., 2023) | 0-shot | 91.8 | 92.9 | 96.5 | 85.1 | 31.7 | - | 64.6 |
| | APRIL-GAN (Chen et al., 2023) | 0-shot | 86.1 | 90.4 | 93.5 | 87.6 | 43.3 | 40.8 | 44.0 |
| | ACR (Li et al., 2023) | 0-shot | 85.8 | 91.3 | 92.9 | 92.5 | 44.2 | 38.9 | 72.7 |
| | MuSc (ours) | 0-shot | **97.8(+6.0)** | **97.5(+4.6)** | **99.1(+2.6)** | **97.3(+4.8)** | **62.6(+18.4)** | **62.7(+21.9)** | **93.8(+21.1)** |
| VisA | WinCLIP (Jeong et al., 2023) | 0-shot | 78.1 | 79.0 | 81.2 | 79.6 | 14.8 | - | 56.8 |
| | APRIL-GAN (Chen et al., 2023) | 0-shot | 78.0 | 78.7 | 81.4 | 94.2 | 32.3 | 25.7 | 86.8 |
| | MuSc (ours) | 0-shot | **92.8(+14.7)** | **89.5(+10.5)** | **93.5(+12.1)** | **98.8(+4.6)** | **48.8(+16.5)** | **45.1(+19.4)** | **92.7(+5.9)** |
| MVTec AD | RegAD (Huang et al., 2022) | 4-shot | 89.1 | 92.4 | 94.9 | 96.2 | 51.7 | 48.3 | 88.0 |
| | PatchCore (Roth et al., 2022) | 4-shot | 88.8±2.6 | 92.6±1.6 | 94.5±1.5 | 94.3±0.5 | 55.0±1.9 | - | 84.3±1.6 |
| | WinCLIP (Jeong et al., 2023) | 4-shot | 95.2±1.3 | 94.7±0.8 | 97.3±0.6 | 96.2±0.3 | 59.5±1.8 | - | 89.0±0.8 |
| | APRIL-GAN (Chen et al., 2023) | 4-shot | 92.8±0.2 | 92.8±0.1 | 96.3±0.1 | 95.9±0.0 | 56.9±0.1 | 54.5±0.2 | 91.8±0.1 |
| | GraphCore (Xie et al., 2023) | 4-shot | 92.9 | - | - | 97.4 | - | - | - |
| | MuSc (ours) | 0-shot | **97.8** | **97.5** | **99.1** | 97.3 | **62.6** | **62.7** | **93.8** |
| VisA | PatchCore (Roth et al., 2022) | 4-shot | 85.3±2.1 | 84.3±1.3 | 87.5±2.1 | 96.8±0.3 | 43.9±3.1 | - | 84.9±1.4 |
| | WinCLIP (Jeong et al., 2023) | 4-shot | 87.3±1.8 | 84.2±1.6 | 88.8±1.8 | 97.2±0.2 | 47.0±3.0 | - | 87.6±0.9 |
| | APRIL-GAN (Chen et al., 2023) | 4-shot | 92.6±0.4 | 88.4±0.5 | **94.5±0.3** | 96.2±0.0 | 40.0±0.4 | 32.2±0.1 | 90.2±0.1 |
| | MuSc (ours) | 0-shot | **92.8** | **89.5** | 93.5 | **98.8** | **48.8** | **45.1** | **92.7** |

(a) MVTec AD                    (b) VisA

Figure 6: Visualization of anomaly segmentation results on MVTec AD and VisA benchmarks.

composed of 6 layers. We extract the patch tokens output from each stage for our method and use the linearly projected class token of the last layer for classification optimization. All the unlabeled test images are scaled to a resolution of 518×518 and then fed into the backbone. In the MSM module, we use the minimum 30% anomaly scores. In the RsCIN module, we use the multi-window setting with 2, 3 on MVTec AD and 8, 9 on VisA.

**Datasets.** To study Industrial anomaly classification and segmentation performance, we conduct experiments on industrial image datasets MVTec AD (Bergmann et al., 2019) and VisA (Zou et al., 2022). MVTec AD has high-resolution (from $700^2$ to $1024^2$) RGB images of 10 object categories and 5 texture categories. VisA comprises high-resolution (1000×1500) RGB images covering 12 objects in 3 domains. Both benchmarks contain normal and anomaly images in the test images.

**Evaluation Metrics.** For classification metrics, we report 3 metrics: the Area Under Receiver Operator Characteristic curve (AUROC), Average Precision (AP), and F1-score at optimal threshold (F1-max). For segmentation metrics, we report 4 metrics: pixel-wise AUROC, pixel-wise F1-max, pixel-wise AP, and Per-Region Overlap (PRO).

**Baselines.** We compare our method with some state-of-the-art zero/few-shot approaches, e.g. WinCLIP (Jeong et al., 2023), APRIL-GAN (Chen et al., 2023), RegAD (Huang et al., 2022), PatchCore (Roth et al., 2022) ,GraphCore (Xie et al., 2023) and ACR Li et al. (2023). Among these methods. APRIL-GAN is in 1st place for the Zero-shot Track of the CVPR 2023 VAND Workshop Challenge, which uses an additional labeled auxiliary dataset for training. RegAD and ACR use other categories of the dataset when training. The unmeasured metrics in these papers are reproduced by using the official implementation code. In addition, we also compare our method with some full-shot methods.

## 4.1 QUANTITATIVE AND QUALITATIVE RESULTS

**Comparison with zero/few-shot methods.** In Tab. 1, we compare MuSc with some state-of-the-art zero-shot methods on MVTec AD and VisA datasets. On MVTec AD, MuSc achieves **21.1%**

Table 2: Comparison with some existing many-shot methods in image-AUROC and pixel-AUROC (%) on MVTec AD.

| Methods | Setting | AC | AS |
|---|---|---|---|
| CutPaste (Li et al., 2021) | full-shot | 95.2 | 88.3 |
| NSA (Schlüter et al., 2022) | full-shot | 97.2 | 96.3 |
| IGD (Chen et al., 2022) | full-shot | 93.4 | 93.0 |
| PatchCore (Roth et al., 2022) | full-shot | 99.6 | 98.2 |
| RegAD (Huang et al., 2022) | 32-shot | 94.6 | 96.9 |
| GraphCore (Xie et al., 2023) | 8-shot | 95.9 | 97.8 |
| MuSc(ours) | 0-shot | 97.8 | 97.3 |

Table 3: Ablation study of LNAMD with different aggregation degrees $r$ in image-AUROC and pixel-AUROC (%) on MVTec AD and VisA.

| | MVTec AD | | VisA | |
|---|---|---|---|---|
| $r$ | AC | AS | AC | AS |
| $\{1\}$ | 96.8 | 94.6 | 93.0 | 97.9 |
| $\{3\}$ | 97.3 | 97.1 | 89.5 | 98.4 |
| $\{5\}$ | 94.5 | 97.6 | 82.2 | 97.6 |
| $\{1,3\}$ | 98.1 | 96.3 | 93.7 | 98.6 |
| $\{3,5\}$ | 96.5 | 97.6 | 87.2 | 98.3 |
| $\{1,3,5\}$ | 97.8 | 97.3 | 92.8 | 98.7 |

Table 4: Ablation study of MSM with different sample strategies in image-AUROC and pixel-AUROC (%) on MVTec AD and VisA.

| | MVTec AD | | VisA | |
|---|---|---|---|---|
| setting | AC | AS | AC | AS |
| (a)min | 96.8 | 94.1 | 91.0 | 98.4 |
| (b)max | 83.8 | 92.9 | 74.9 | 95.7 |
| (c)mean | 95.0 | 96.8 | 89.6 | 98.4 |
| (d)30% + min | 96.8 | 94.1 | 91.0 | 98.4 |
| (e)30% + max | 96.0 | **97.5** | 91.2 | 98.6 |
| (f)30% + mean | **97.8** | 97.3 | **92.8** | **98.7** |

Table 5: Ablation study of RsCIN. We report anomaly classification performance in AUROC, F1-max and AP on MVTec AD and VisA dataset. All metrics are in %. Bold indicates the best performance.

| Datasets | RsCIN | AUROC | F1-max | AP |
|---|---|---|---|---|
| MVTec AD | w/o | 97.4 | 96.6 | 98.7 |
| | w | **97.8** | **97.5** | **99.1** |
| VisA | w/o | 90 | 87.1 | 90.9 |
| | w | **92.8** | **89.5** | **93.5** |

improvement on PRO and **21.9%** improvement on AP than the second-best method in the zero-shot setting. On VisA, in classification and segmentation, our method has more than **10%** improvement compared with the second-best method. By comparing with some 4-shot methods, the classification and segmentation results of MuSc beyond most methods with four normal reference images, although we do not use any labeled images.

We show the industrial image visualization of anomaly maps of MuSc in Fig. 6. Compared with other zero/few-shot methods, our method can detect small defects and has fewer false positives, as shown in the *pill* and *cashew*. Other methods in zero/few-shot are sensitive to the normal spots on the *pill* and the tiny splinters on the *cashew*. To the large defects shown in *bottle* and *pcb1*, our method can segment the whole regions. Some logical anomalies, such as the flip of *metalnut* and the proliferation of *pipe_fryum*, can be segmented entirely by our method.

**Comparison with many-shot methods.** We compare our MuSc with some many-shot methods on MVTec AD, shown in Tab. 2. MuSc is comparable to some full-shot methods, such as CutPaste (Li et al., 2021), IGD (Chen et al., 2022) and NSA (Schlüter et al., 2022), which utilize all the normal reference images in the train set. Without using any labeled images, our method also outperforms RegAD with 32-shot and is competitive with GraphCore with 8-shot.

## 4.2 ABLATION STUDY

**The influence of the aggregation degree $r$.** The aggregation degree $r$ controls the size of the local neighborhood to be aggregated in the LNAMD module. As reported in Tab. 3, we conduct the experiments with combinations of different aggregation degrees. The experimental results demonstrate that using all three aggregation degrees of $r \in \{1, 3, 5\}$ is better than other cases in the comprehensive results of AC and AS. Moreover, due to the different datasets with varying anomaly sizes, a smaller $r$ is suitable for segmenting small abnormal regions, which is common on VisA, while a larger $r$ does well in segmenting large abnormal regions, which is common on MVTec AD. To balance the representation on different datasets, we use three scales of aggregation degrees.

**Discussion of the mutual scoring mechanism.** Here, we modify Eq. 2 in six different cases, shown in Tab. 4. In (a), (b), and (c), we remove the operation of selecting the minimum 30% value interval. (a) and (d) means replace the average operation with the minimum operation, and in (b) and (e) we replace it with the maximum operation. (f) is the optimal setting for our method. We also test the effect of the selection percentage shown in Fig. 7. Selecting the minimum 30% can bring better comprehensive AC and AS results on the two datasets.

Table 6: Per-image inference time and maximum GPU memory cost by our MuSc. We divide the whole dataset into $s$ subsets, measured at NVIDIA RTX 3090 GPU. We report the time and GPU memory cost with different $s$.

| $s$ | Time (ms) | GPU cost (MB) |
|---|---|---|
| 1 | 998.8 | 7168 |
| 2 | 605.8 | 5666 |
| 3 | 513.5 | 5026 |

Table 7: The effect of dataset size. We divide the dataset into $s$ parts, evaluate image-AUROC and pixel-AUROC(%) on MVTec AD and VisA.

| Datasets | $s$ | AC | AS | Size of the subsets |
|---|---|---|---|---|
| MVTec AD | 1 | 97.8 | 97.3 | 42-167 |
| | 2 | 97.1 | 97.2 | 21-84 |
| | 3 | 96.7 | 97.3 | 14-56 |
| VisA | 1 | 92.8 | 98.8 | 150-200 |
| | 2 | 92.5 | 98.7 | 75-100 |
| | 3 | 92.3 | 98.6 | 50-67 |

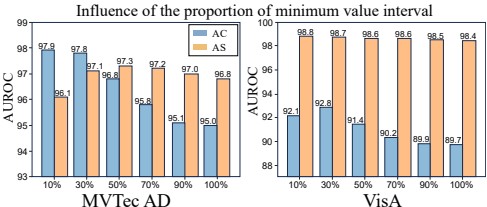

Figure 7: The influence of the proportion of minimum value interval selection in MSM.

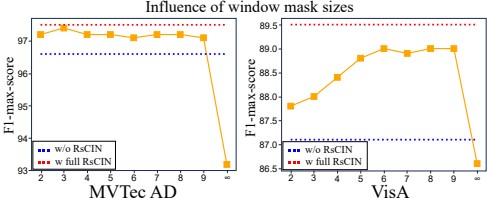

Figure 8: The influence of different window mask sizes on MVTec AD and VisA.

**Discussion of the re-scoring with constrained image-level neighborhood.** In Tab. 5, we remove the re-scoring with constrained image-level neighborhood module to check our method for classification. The results demonstrate the RsCIN module could bring $0.9\%$ F1-max-score gains on MVTec AD and $2.8\%$ AUROC gains on VisA. Meanwhile, for the multi-window mask operation of RsCIN, we replace the multi-window with the single-window $k \in \{2, ..., 9\}$ and full-window $k = \infty$ without the mask operation in Fig. 8. Our multi-window strategy outperforms the single-scale strategy on both datasets. In addition, we insert our RsCIN module into some other different AC/AS methods, and the classification results have been improved, detailed in Appendix A.2.4.

**Discussion of inference time and memory cost.** The inference time and memory cost are also what we are concerned about. In our MuSc, we report the per-image inference time and maximum GPU memory cost in Tab. 6, where the memory cost is measured on the most numerous dataset which contains 200 images. When the number of test images increases, the inference time and memory cost continue to increase, which may become a limitation of our method.

To break through this limitation, we divide the entire test images into $s$ subsets, calculate the anomaly scores on every subset, and finally compute the AC and AS metrics together. The AC/AS results are shown in Tab. 7, where *42-167* means the maximum size of the divided subset is *167* and the minimum size is *42*. As the subset size decreases, the performance decreases by less than $1.1\%$ for AC and $0.2\%$ for AS, which means our method works well even with tiny datasets. The inference time and memory cost also decrease as the subset size decreases, shown in Tab. 6. Therefore, when the test set is large, we consider using our MuSc in its subsets separately. Additionally, we conduct a comprehensive efficiency comparison in Appendix A.4.

**Discussion of few-shot setting.** Our MuSc is a pure zero-shot method that does not require any labeled images. We design two extension methods in the few-shot setting based on our method when a small number of normal reference images are provided, see Appendix A.6. There is no significant improvement in the results when a few images are added because our method already uses a large amount of normal prior information in the unlabeled test images.

## 5 CONCLUSION

In this paper, we propose a novel zero-shot industrial AC and AS framework named MuSc. We explore the normal and abnormal cues implicit in the unlabeled test images. First, we fuse local neighborhood aggregated patch tokens in different degrees. Then, we use a mutual scoring mechanism for each image. Finally, we propose an image-level AC optimization method. Notably, our method surpasses the existing zero-shot approaches, even outperforms most of the few-shot approaches, and is comparable to some full-shot methods.

## 6 ACKNOWLEDGMENTS

This work was supported by the National Natural Science Foundation of China under Grant No. 62176098. The computation is completed in the HPC Platform of Huazhong University of Science and Technology.

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

# A APPENDIX

## A.1 METHOD DETAILS

### A.1.1 MSM MODULE DETAILS

The intuition of MSM is based on the observation of industrial product images, the normal image patches could find a relatively large number of similar patches in other unlabeled images, while the abnormal ones only have a few similar patches. In order to clearly explain the detailed theories of the MSM module, we divide it into three steps.

The first step is to score each patch of each image using the other unlabeled images. We perform LNAMD on the patch tokens of ViT, each aggregated patch token is denoted as $\hat{p}_{i,l}^{m,r} \in \mathbb{R}^{1 \times C}$, where $m \in [1, M]$ indicates the index of the patch, $r$ is the aggregation degree and $l \in \{1, 2, ..., L\}$ is the stage $l$ of ViT. For a given aggregation degree $r$ and stage $l$ of ViT, if the patch token $\hat{p}_{i,l}^{m,r}$ of $I_i$ is similar to any patch token $\hat{p}_{j,l}^{n,r}$ of $I_j$, the image $I_j$ assigns a small anomaly score to $\hat{p}_{i,l}^{m,r}$, see Eq. 1. Specifically, we measure the Euclidean distance between $\hat{p}_{i,l}^{m,r}$ and all patches in $I_j$. The $\min$ operator is used to find the most similar patch in $I_j$ and uses it to score $\hat{p}_{i,l}^{m,r}$. When $\hat{p}_{i,l}^{m,r}$ represents a normal region, the most similar patch in $I_j$ is usually a normal image patch, resulting in a relatively small anomaly score. While $\hat{p}_{i,l}^{m,r}$ represents an anomalous region, it is difficult to find a similar patch in $I_j$, resulting in a larger anomaly score.

The second step is to selectively utilize the anomaly scores derived from unlabeled images. We propose an Interval Average (IA) operation, which only utilizes the minimum $X\%$ of the scoring vector $A_{i,l}^{m,r}$ as defined earlier. In Tab. 4, compared to the common practice of averaging all images, using IA to select the lowest 30% scores could improve the image-AUROC by 2.8% and the pixel-AUROC by 0.5% on MVTec AD. Regarding the overlap in Fig. 4(c), we observe that it is caused by normal patches with appearance variations from different normal images. These dissimilar normal patches are assigned higher anomaly scores by other normal patches. Consequently, for patches representing a normal region, the IA operation can often find similar normal patches in some unlabeled images, thereby reducing the anomaly scores of normal patches. For abnormal patches, due to the randomness and unpredictability of the anomalies, the IA operation cannot find similar patches in unlabeled images. It also maintains a higher abnormal score for the abnormal patch, further widening the gap between normal and abnormal patches.

The third step involves combining anomaly scores from multiple degrees and stages. We calculate Eq. 1 and Eq. 2 for each combination of stage $l$ and aggregation degree $r$ respectively. Then we calculate the average of all $\overline{a}_{i,l}^{m,r}$ to obtain the patch-level anomaly score using Eq. 3.

### A.1.2 RSCIN MODULE DETAILS

In order to clearly explain the detailed theory of RsCIN module, we take it to optimize the pixel-level anomaly classification score $c_i$ of image $I_i$ as an example. We assume that in the Multi-window Mask Operation, $k \in \{k_1, k_2\}$, where $k_1 < k_2$. According to the similarity matrix $W$, we define the $k$-nearest neighbor to image $I_i$ as $\{\hat{I}_1, ..., \hat{I}_k\}$. Then their similarities to image $I_i$ are defined as $\{w_{i,1}, ..., w_{i,k}\}$, which are obtained by $M_k \odot W$ in Eq. 5. The $D^{-1}$ is used to make the sum of these similarities equal to 1. The results of the transformation over different $k$-nearest neighbors are defined as $\{\hat{w}_{i,1}^{k_1}, ..., \hat{w}_{i,k_1}^{k_1}\}$ and $\{\hat{w}_{i,1}^{k_2}, ..., \hat{w}_{i,k_2}^{k_2}\}$. The equations for calculating $\hat{w}_{i,j}^{k_1}$ and $\hat{w}_{i,j}^{k_2}$ are,

$$
\begin{aligned}
\hat{w}_{i,j}^{k_1} &= \frac{w_{i,j}}{w_{i,1} + w_{i,2} + ... + w_{i,k_1}}, \quad \text{where } \hat{w}_{i,j}^{k_1} \in \{\hat{w}_{i,1}^{k_1}, ..., \hat{w}_{i,k_1}^{k_1}\} \\
\hat{w}_{i,j}^{k_2} &= \frac{w_{i,j}}{w_{i,1} + w_{i,2} + ... + w_{i,k_2}}, \quad \text{where } \hat{w}_{i,j}^{k_2} \in \{\hat{w}_{i,1}^{k_2}, ..., \hat{w}_{i,k_2}^{k_2}\}
\end{aligned} \tag{6}
$$

where $\hat{w}_{i,j}^{k}$ means the normalized similarity between image $I_i$ and image $I_j$ in the $k$-nearest neighbor, $k \in \{k_1, k_2\}$. The pixel-level anomaly classification scores of the $k$ images $\{\hat{I}_1, ..., \hat{I}_k\}$ are $\{\overline{c}_1, ..., \overline{c}_k\}$. Then using Eq. 5 to optimize only the anomaly classification score $c_i$ of image $I_i$ can be rewritten as,

$$
\begin{aligned}
\hat{c}_i &= (\sum_{k \in \{k_1, k_2\}} (\hat{w}_{i,1}^k \overline{c}_1 + ... + \hat{w}_{i,k}^k \overline{c}_k) + c_i)/(2+1) \\
&= \frac{c_i}{3} + \frac{1}{3} \sum_{k \in \{k_1, k_2\}} (\hat{w}_{i,1}^k \overline{c}_1 + ... + \hat{w}_{i,k}^k \overline{c}_k) \\
&= \frac{c_i}{3} + \frac{1}{3}(\hat{w}_{i,1}^{k_1} \overline{c}_1 + ... + \hat{w}_{i,k_1}^{k_1} \overline{c}_{k_1} + \hat{w}_{i,1}^{k_2} \overline{c}_1 + ... + \hat{w}_{i,k_2}^{k_2} \overline{c}_{k_2}) \\
&= \frac{c_i}{3} + \frac{1}{3}(\hat{w}_{i,1}^{k_1} \overline{c}_1 + ... + \hat{w}_{i,k_1}^{k_1} \overline{c}_{k_1} + \hat{w}_{i,1}^{k_2} \overline{c}_1 + ... + \hat{w}_{i,k_1}^{k_2} \overline{c}_{k_1} + \hat{w}_{i,k_1+1}^{k_2} \overline{c}_{k_1+1} + ... + \hat{w}_{i,k_2}^{k_2} \overline{c}_{k_2}) \\
&= \frac{c_i}{3} + \frac{1}{3}((\hat{w}_{i,1}^{k_1} + \hat{w}_{i,1}^{k_2})\overline{c}_1 + ... + (\hat{w}_{i,k_1}^{k_1} + \hat{w}_{i,k_1}^{k_2})\overline{c}_{k_1} + \hat{w}_{i,k_1+1}^{k_2} \overline{c}_{k_1+1} + ... + \hat{w}_{i,k_2}^{k_2} \overline{c}_{k_2}) \\
&= \frac{c_i}{3} + \frac{1}{3} \sum_{j=1}^{k_2} \overline{w}_{i,j} \overline{c}_j
\end{aligned}
\tag{7}
$$

where $\hat{c}_i$ represents the optimized anomaly classification score of image $I_i$, $\overline{w}_{i,j}$ is the similarity between image $I_i$ and image $I_j$ under the multi-window setting,

$$
\overline{w}_{i,j} = \begin{cases} \hat{w}_{i,j}^{k_1} + \hat{w}_{i,j}^{k_2}, & \text{if } 0 < j \leq k_1 \\ \hat{w}_{i,j}^{k_2}, & \text{if } k_1 < j \leq k_2 \end{cases}
\tag{8}
$$

Eq. 7 shows that $\hat{c}_i$ is affected by anomaly classification scores in multiple $k$-nearest neighbors. According to this equation, the value of $c_i$ increases if image $I_i$ has high-scoring $k$-neighbors (i.e., scores $\overline{c}_j \in \{\overline{c}_1, ..., \overline{c}_k\}$ are high), and vice versa.

## A.2 ADDITIONAL ABLATION RESULTS

### A.2.1 EFFECT OF DIFFERENT BACKBONES

Table 8: Comparison of our MuSc performance in image-AUROC and pixel-AUROC (%) on MVTec AD and VisA benchmarks across different pre-training methods with various backbone architectures.

| Pre-training Method | Arch. | Pre-training Dataset | MVTec AD | | VisA | |
|---|---|---|---|---|---|---|
| | | | AC | AS | AC | AS |
| DINO | ViT-B-16 | ImageNet-1k | 94.9 | 97.8 | 88.2 | 97.8 |
| | ViT-B-8 | ImageNet-1k | 96.2 | 98.1 | 93.5 | 98.2 |
| DINOv2 | ViT-B-14 | LVD-142M | 96.4 | 98.0 | 89.7 | 98.4 |
| | ViT-L-14 | LVD-142M | 97.1 | 98.1 | 90.5 | 98.4 |
| CLIP | ViT-B-32 | WIT-400M | 90.0 | 95.6 | 79.3 | 95.8 |
| | ViT-B-16 | WIT-400M | 94.8 | 96.9 | 85.7 | 97.6 |
| | ViT-B-16-plus-240 | LAION-400M | 95.4 | 97.1 | 89.3 | 97.7 |
| | ViT-L-14 | WIT-400M | 96.1 | 97.4 | 90.9 | 98.5 |
| | ViT-L-14-336 | WIT-400M | 97.8 | 97.3 | 92.8 | 98.8 |
| - | Swin-B-4 | ImageNet-22k | 87.7 | 92.9 | 80.1 | 86.2 |
| | Swin-L-4 | ImageNet-22k | 87.6 | 90.9 | 76.7 | 82.1 |

We explore the effect of different pre-training methods with various backbone architectures in our method, such as DINO (Caron et al., 2021), DINOv2 (Oquab et al., 2023), and CLIP (Radford et al., 2021). In addition, we also conduct experiments on the swin transformer (Liu et al., 2021), which uses varied-size window attention to calculate attention inside local windows to detect objects with different sizes. ViT-Large contains 24 layers, we keep the same setup with our experiment setting to divide them into 4 stages, and each stage comprises 6 layers. ViT-Base contains 12 layers, we also

divide them evenly into 4 stages, and each stage comprises 3 layers. For the swin transformer, we follow its own setting to divide the whole backbone into 4 stages. Then we extract the patch tokens output from each stage for our method and use the linearly projected class token of the last layer for classification optimization, except the swin transformer without the class token. All other settings remain the same as those in the main text.

We show the image-AUROC and pixel-AUROC using different pre-training methods with various backbone architectures in Tab. 8. CLIP with ViT-L-14-336 achieves better comprehensive results in anomaly classification and segmentation on MVTec AD and VisA datasets. We notice that the larger model with high resolution and small patch size is more suitable for AC and AS. Due to the large patches in some stages of swin transformer, the patch-level features extracted by it do not work well under our method.

### A.2.2 ADDITIONAL LNAMD ABLATION RESULTS

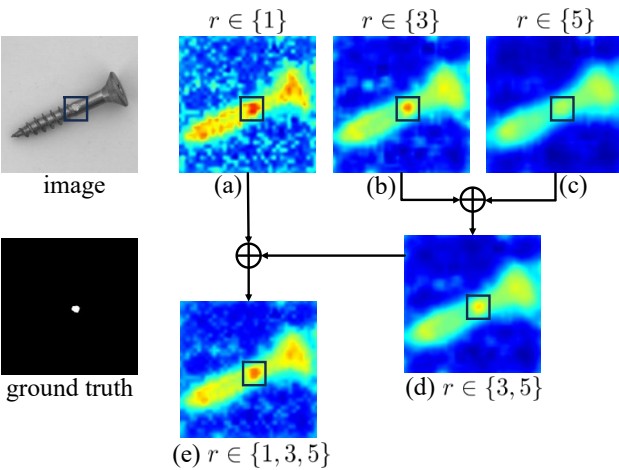

Figure 9: Visualization of the combination results of different aggregation degrees $r$.

We further study the implications of combining different aggregation degrees $r \in \{1, 3, 5\}$, and visualize the results of these combinations in Fig. 9. The defect in the screw is relatively small in this example. The small defect is smoothed and has a lower score in large aggregation degree ($r \in \{5\}$). We add the anomaly score with aggregation degree ($r \in \{5\}$) to the anomaly score with aggregation degree ($r \in \{3\}$). We can see from (d) that the combination of ($r \in \{3, 5\}$) results in a smaller anomaly score being added to a larger anomaly score in the small anomaly region. This leads to a decrease in the anomaly score and false negative. When we add a larger anomaly score with aggregation degree ($r \in \{1\}$), the anomaly score within the black box in (e) increases and the false-negative reduces. In order to take into account the various sizes of anomalies in real industrial scenarios, we use ($r \in \{1, 3, 5\}$) to maintain balance.

Table 9: Ablation study of LNAMD with different aggregation degrees $r$ in image-AUROC and pixel-AUROC (%) on MVTec AD and VisA benchmarks.

| $r$ | MVTec AD | | VisA | |
|---|---|---|---|---|
| | AC | AS | AC | AS |
| $\{1\}$ | 96.9 | 94.6 | 92.8 | 97.9 |
| $\{1, 3\}$ | 97.7 | 96.3 | 93.7 | 98.6 |
| $\{1, 3, 5\}$ | 97.8 | 97.3 | 92.8 | 98.7 |
| $\{1, 3, 5, 7\}$ | 97.3 | 97.5 | 90.8 | 98.7 |
| $\{1, 3, 5, 7, 9\}$ | 96.4 | 97.8 | 89.5 | 98.6 |
| $\{1, 3, 5, 7, 9, 13\}$ | 95.7 | 97.9 | 88.4 | 98.5 |

We explore the results of using more aggregation degrees $r$ in the LNAMD module, shown in Tab. 9. With the addition of a larger aggregation degree, we observe that the classification results continued to decline, and the effect on segmentation is related to the size of the anomalies in the two datasets.

There are more large anomalies in MVTec AD, so a larger aggregation degree is conducive to the segmentation of large anomalies, while most of the anomalies in VisA are small, and the use of a smaller aggregation degree has greater advantages.

### A.2.3 ADDITIONAL MSM RESULTS

In the MSM module, we observe that some normal patches can only find a small amount of similar patches due to the appearance variations of different normal images. To reduce these dissimilar patches in scoring, we perform an Interval Average operation on the minimum 30% of $A_{i,m}^{l,r}$. In this process, a small number of patches that are similar to abnormal patches are retained, shown in the red dotted box at the left corner of Fig. 10, which is the same as Fig. 4. This part will make the anomaly scores of some abnormal patches smaller, which may lead to false negatives. Therefore, we conduct the experiment to perform the interval average operation on the minimum $Y\%$ to 30% of $A_{i,m}^{l,r}$ instead of only 30%. Results are shown in Tab. 10, as the minimum $Y\%$ are removed, the classification and segmentation results get worse. For the unlabeled test set, directly removing the minimum part of anomaly scores will increase the anomaly scores of some normal patches, which may lead to some false positive cases, and the impact caused by this part will be greater than false negative cases caused by abnormal patches on the two datasets.

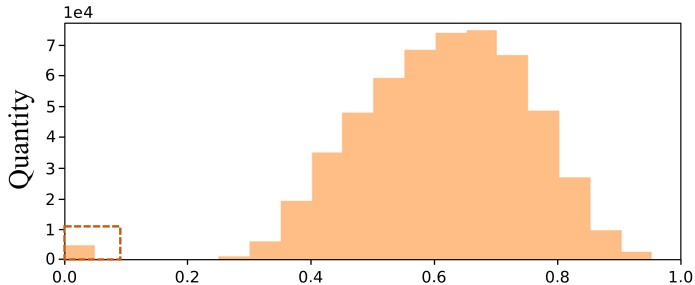

Figure 10: The histogram of anomaly score of abnormal patch token.

Table 10: Ablation study of MSM with the different percentage interval selection in image-AUROC and pixel-AUROC (%) on MVTec AD and VisA.

| percentage interval | MVTec AD | | VisA | |
|---|---|---|---|---|
| | AC | AS | AC | AS |
| $0\% \sim 30\%$ | 97.8 | 97.3 | 92.5 | 98.7 |
| $2\% \sim 30\%$ | 97.7 | 97.2 | 92.5 | 98.7 |
| $4\% \sim 30\%$ | 97.5 | 97.2 | 92.4 | 98.7 |
| $6\% \sim 30\%$ | 97.3 | 97.3 | 92.3 | 98.7 |
| $8\% \sim 30\%$ | 97.3 | 97.3 | 92.2 | 98.7 |
| $10\% \sim 30\%$ | 97.1 | 97.4 | 92.0 | 98.7 |

### A.2.4 ADDITIONAL RSCIN RESULTS

In Tab. 11, we verify that our RsCIN module can be considered a general plug-and-play module in the industrial anomaly classification task, which works well in the classification optimization among different anomaly classification methods. We choose some representative methods for each type of anomaly classification. Reconstruction-based approaches such as DRAEM (Zavrtanik et al., 2021) and DSR (Zavrtanik et al., 2022). Methods based on student-teacher architecture such as STPM (Wang et al., 2021). And methods for using memory bank such as SPADE (Cohen & Hoshen, 2020) and PatchCore (Roth et al., 2022). In addition, we also conduct experiments on some zero-shot and few-shot methods, such as RegAD (Huang et al., 2022) in the 2, 4, and 8-shot settings, APRIL-GAN (Chen et al., 2023) in the 0-shot and 4-shot settings.

We insert our RsCIN module at the end of these methods to optimize the classification results. In all experiments, we maintain the same parameter settings, and uniformly use the class token extracted from ViT-L-14-336 as image-level features and input them into the RsCIN module. We observe

Table 11: Ablation study of RsCIN. We report anomaly classification (AC) performance in AUROC, F1-max, and AP. All the methods are tested on MVTec AD except the method marked with an asterisk(*) represents testing on the VisA benchmark. All metrics are in %.

| Method | RsCIN | AUROC | F1-max | AP | Method | RsCIN | AUROC | F1-max | AP |
|---|---|---|---|---|---|---|---|---|---|
| SPADE | w/o | 85.4 | 90.1 | 93.6 | PatchCore | w/o | 99.0 | 98.4 | 99.7 |
| (Cohen & Hoshen, 2020) | w | 87.0 | 91.4 | 94.3 | (Roth et al., 2022) | w | 99.1 | 98.4 | 99.7 |
| DRAEM | w/o | 98.0 | 97.0 | 99.0 | DSR | w/o | 98.2 | 96.6 | 99.1 |
| (Zavrtanik et al., 2021) | w | 97.9 | 97.0 | 99.1 | (Zavrtanik et al., 2022) | w | 98.2 | 96.8 | 99.3 |
| STPM | w/o | 94.9 | 95.8 | 98.2 | RegAD(2-shot) | w/o | 84.8 | 90.7 | 92.5 |
| (Wang et al., 2021) | w | 95.6 | 96.5 | 98.5 | (Huang et al., 2022) | w | 86.2 | 91.6 | 93.1 |
| APRIL-GAN(0-shot) | w/o | 86.1 | 90.4 | 93.5 | RegAD(4-shot) | w/o | 89.1 | 92.4 | 94.9 |
| (Chen et al., 2023) | w | 86.1 | 90.8 | 93.7 | | w | 91.0 | 93.5 | 95.8 |
| APRIL-GAN(4-shot) | w/o | 92.8 | 92.8 | 96.3 | RegAD(8-shot) | w/o | 91.2 | 92.9 | 95.7 |
| | w | 93.4 | 93.1 | 96.8 | | w | 92.1 | 94.0 | 96.0 |
| APRIL-GAN$^*$(0-shot) | w/o | 78.0 | 78.7 | 81.4 | APRIL-GAN$^*$(4-shot) | w/o | 92.6 | 88.4 | 94.5 |
| | w | 78.7 | 80.1 | 82.0 | | w | 94.5 | 90.5 | 95.8 |

that our RsCIN module can improve the classification results of different AC and AS methods. RsCIN significantly improves SPADE, DRAEM, RegAD, and other methods. However, one of the limitations of RsCIN is that for some one-class methods that do not contain image-level features in their backbones, another backbone is needed to extract image-level features. Another limitation is that for methods with very high classification results, the improvement brought by RsCIN is limited by the representations of image-level features.

## A.3 THE INFLUENCE OF ORIENTATIONS AND SCALES

Table 12: Comparison with some zero-shot methods in image-AUROC (AC) and pixel-AUROC (AS) on the categories of MVTec AD and VisA with inconsistent orientations and scales.

| Category | WinCLIP | | APRIL-GAN | | MuSc | |
|---|---|---|---|---|---|---|
| | AC | AS | AC | AS | AC | AS |
| screw | 83.3 | 89.6 | 84.9 | 97.8 | 83.5 | 98.9 |
| hazelnut | 93.9 | 94.3 | 89.6 | 96.1 | 99.6 | 99.4 |
| metal_nut | 97.1 | 61.0 | 68.4 | 65.4 | 96.3 | 86.0 |
| capsules | 85.0 | 81.6 | 61.2 | 97.5 | 88.8 | 98.8 |
| macaroni2 | 63.7 | 59.3 | 64.6 | 97.8 | 69.9 | 97.2 |
| mean | 84.6 | 77.2 | 73.7 | 90.9 | 87.6 | 96.1 |
| mean-ALL | 91.8 | 85.1 | 86.1 | 87.6 | 97.8 | 97.3 |

We select the categories with inconsistent orientations or scales from MVTec AD and VisA datasets for the additional experimental evaluation, which is given in Tab. 12. We compare our method with WinCLIP and APRIL-GAN and present the image-AUROC/pixel-AUROC in this table. We observe that both our approach and WinCLIP are influenced by inconsistent orientations or scales. we find the reason is that both of us use the fixed pre-trained vision transformer as the feature extractor, and as introduced in (Radford et al., 2021), during the pre-training process, the vision transformer only performs minor data augmentation on orientations and scales. Therefore, neither of us can effectively address the inconsistent orientations or scales. APRIL-GAN also achieves a decreased AC score, while it obtaining a better AS score. We guess the reason is that APRIL-GAN utilizes an additional training set to optimize the AS. In addition, our approach still outperforms WinCLIP and APRIL-GAN in case of inconsistent orientations and scales.

## A.4 DETAILED COMPARISON OF INFERENCE TIME AND MEMORY COST

In Tab. 13, we show the inference time and GPU memory cost of our method under different settings. We also compare them with other zero/few-shot methods. WinCLIP and APRIL-GAN are in zero-shot setting, and RegAD is in 4-shot setting. In order to maintain the same backbone as these methods, we use both ViT-L-14-336 and ViT-B-16-plus-240. We find that the replacement

Table 13: Comparison with some zero/few-shot methods in per-image inference time (ms), GPU cost (MB), image-AUROC (%) and AUPRO (%) on MVTec AD.

| Method | Backbone | Training | time(ms) | GPU(MB) | AC | AS |
|---|---|---|---|---|---|---|
| RegAD (Huang et al., 2022) | ResNet-18 | yes | 309.4 | 8920 | 89.1 | 88.0 |
| APRIL-GAN (Chen et al., 2023) | ViT-L-14-336 | yes | 100.2 | 4996 | 86.1 | 44.0 |
| MuSc($s = 1$) | ViT-L-14-336 | no | 998.8 | 7168 | 97.8 | 93.8 |
| MuSc($s = 2$) | ViT-L-14-336 | no | 605.8 | 5666 | 97.1 | 93.8 |
| MuSc($s = 3$) | ViT-L-14-336 | no | 513.5 | 5026 | 96.7 | 93.7 |
| APRIL-GAN | ViT-B-16-plus-240 | yes | 64.7 | 3226 | - | - |
| WinCLIP (Jeong et al., 2023) | ViT-B-16-plus-240 | no | 389 | - | 91.8 | 64.6 |
| MuSc($s = 1$) | ViT-B-16-plus-240 | no | 455.3 | 3002 | 95.4 | 91.9 |
| MuSc($s = 2$) | ViT-B-16-plus-240 | no | 285.7 | 2740 | 95.4 | 91.5 |
| MuSc($s = 3$) | ViT-B-16-plus-240 | no | 238.3 | 2684 | 95.2 | 91.4 |

of ViT-L-14-336 with ViT-B-16-plus-240 resulted in a significant reduction in inference time and GPU memory cost. When we use ViT-L-14-336 as the feature extractor, the large ViT divides the original image into more patches. In the LNAMD module, all patches are required to perform local neighborhood aggregation, and the time increases as the number of patches increases. These patches are aggregated with different aggregation degrees $r \in \{1, 3, 5\}$, and saved for later use. The aggregated features in the LNAMD module consume a significant amount of memory. In the MSM module, mutual scoring is performed on each combination of stage $l$ and aggregation degree $r$, the patch $m$ of image $I_i$ needs to calculate the distances with all patches of all other unlabeled images. The size of unlabeled images is closely related to the inference time of the MSM module.

For our method, we divide the entire dataset of unlabeled images into $s$ subsets in the same way as the ablation study, which is highly effective in improving the inference time and reducing GPU memory cost. Under the setting of using ViT-L-14-336 as the feature extractor, the inference time of our method is $513.5ms$ when $s = 3$, which is half the time consumed when using the entire unlabeled set. Under the setting of ViT-B-16-plus-240, our inference time is further reduced to $238.3ms$ and the GPU memory cost is only $2684MB$, which is $150.7ms$ less than that of the WinCLIP method. When we divide the original dataset into three subsets, the segmentation performance reduces by less than $0.1\%$, and the classification result decreases by $1.1\%$ with ViT-L-14-336. When we use ViT-B-16-plus-240 as our backbone, the segmentation performance reduces by less than $0.5\%$, and the classification result decreases by $0.2\%$. As the subset size decreases, the classification and segmentation performance decreases very little, which proves that our method also achieves good results on the small-scale set of unlabeled images.

Table 14: Comparison with some existing zero/many-shot methods in image-AUROC and pixel-AUROC (%) on **BTAD** dataset.

| Method | Setting | AC | AS |
|---|---|---|---|
| VT-ADL (Mishra et al., 2021) | full-shot | 83.7 | 90.0 |
| P-SVDD (Yi & Yoon, 2020) | full-shot | 83.3 | 92.1 |
| SPADE (Cohen & Hoshen, 2020) | full-shot | 87.6 | 96.9 |
| PaDIM (Defard et al., 2021) | full-shot | 93.7 | 97.3 |
| PyramidFlow (Lei et al., 2023) | full-shot | 95.8 | 97.7 |
| PatchCore (Roth et al., 2022) | 4-shot | 91.4 | 96.3 |
| RegAD (Huang et al., 2022) | 4-shot | 91.0 | 97.5 |
| APRIL-GAN (Chen et al., 2023) | 4-shot | 91.3 | 92.4 |
| APRIL-GAN | 0-shot | 72.3 | 89.6 |
| MuSc | 0-shot | 94.8 | 97.3 |

## A.5 ADDITIONAL DATASET RESULTS

To demonstrate the stability of the hyperparameters in MuSc, we directly applied the hyperparameters on the MVTec dataset to test on additional datasets. In Tab. 14, we compare our MuSc with

Table 15: k-shot results of MuSc in image-AUROC and pixel-AUROC (%) on **MVTec AD**.

| setting | AC | AS |
|---------|------|------|
| 0-shot | 97.8 | 97.3 |
| 1-shot | 97.8 | 97.3 |
| 2-shot | 97.7 | 97.3 |
| 4-shot | 97.8 | 97.4 |
| 8-shot | 97.8 | 97.4 |
| 16-shot | 97.9 | 97.5 |
| 32-shot | 98.0 | 97.4 |
| full-shot | 98.4 | 97.9 |

Table 16: k-shot results of MuSc+ in image-AUROC and pixel-AUROC (%) on **MVTec AD**.

| setting | AC | AS |
|---------|------|------|
| 0-shot(MuSc) | 97.8 | 97.3 |
| 1-shot | 92.7 | 97.2 |
| 2-shot | 93.5 | 97.4 |
| 4-shot | 96.5 | 97.8 |
| 8-shot | 97.3 | 98.1 |
| 16-shot | 98.0 | 98.3 |
| 32-shot | 98.7 | 98.4 |
| full-shot | 99.5 | 98.5 |

several many-shot methods on the BTAD dataset. In zero-shot setting, MuSc achieves a 22.5% image-AUROC gain and a 7.7 % pixel-AUROC gain compared to APRIL-GAN. MuSc even outperforms RegAD and PatchCore in 4-shot setting and is competitive with some methods in full-shot setting. We emphasize that MuSc achieves excellent results even without adjusting the hyperparameters for a specific dataset.

## A.6 RESULTS OF TWO EXTENSION METHODS IN FEW/MANY-SHOT SETTING

In the few/many-shot setting, several normal reference images $D_{\text{ref}} = \{I_i, i = 1, ..., R\}$ are randomly drawn from labeled normal train set $D_{\text{train}}$. Keeping a similar structure to MuSc, we designed two few/many-shot extension methods. In the first extension, we use the same structure as the original MuSc and input the $D_{\text{ref}}$. In the second extension, we design the MuSc+ with minor modifications to MuSc to make better use of labeled normal images.

### A.6.1 MUSC FOR FEW-SHOT

Keep the MuSc method, we simply add these reference images $D_{\text{ref}}$ in MSM module. Given a test image $I_i$, when using other images to calculate the anomaly score for each aggregated patch token of $I_i$ following Eq. 1, we not only use the images in $\{D_u \backslash I_i\}$, but also use these given normal reference images in $D_{\text{ref}}$. Then we follow Eq. 2 and Eq. 3 to obtain few/many-shot segmentation results and get classification results by RsCIN in Section 3.3. The results are shown in Tab. 15. We notice that some normal reference images are added resulting in a small improvement because we have used a large amount of normal implicit information from unlabeled test images. The information is limited when a small number of normal images are added.

### A.6.2 MUSC+

In our MuSc, some abnormal images also participate in mutual scoring, which is unavoidable in unlabeled test images. Our MuSc is designed with this case, so when some labeled normal images are added, we design a different scoring mechanism to process these images. By changing the MSM module in our MuSc, we propose a new scoring mechanism to obtain the anomaly score by only using the normal reference images $D_{\text{ref}}$. Given a test image $I_i$, we leverage the images in $D_{\text{ref}}$ to assign the anomaly score for each aggregated patch token of $I_i$ in layer $l$ using aggregation degree $r$, following Eq. 9,

$$\tilde{a}_{i,l}^{m,r}(I_j) = \min_n \|\hat{p}_{i,l}^{m,r} - \hat{p}_{j,l}^{n,r}\|_2 \tag{9}$$

where $I_j$ indicates that image $I_j \in \{D_{\text{ref}}\}$ is employed for scoring, and hence we obtain the scoring vector $A_{i,l}^{m,r} = [a_{i,l}^{m,r}(I_1), a_{i,l}^{m,r}(I_2), ..., a_{i,l}^{m,r}(I_R)]$ for each $l$ and $r$ according to $D_{\text{ref}}$. We perform a minimization operation on $A_{i,l}^{m,r}$, i.e.,

$$\tilde{a}_{i,l}^{m,r} = \min_j \tilde{a}_{i,l}^{m,r}(I_j) \tag{10}$$

we perform Eq. 9 and Eq. 10 on each combination of layer $l$ and aggregation degree $r$, respectively, and compute the average of all $\tilde{a}_{i,l}^{m,r}$ to achieve the patch-level anomaly score $\tilde{\mathbf{a}}_i^m$.

$$\tilde{\mathbf{a}}_i^m = \frac{1}{L} \sum_{l \in \{1,...,L\}} \frac{1}{3} \sum_{r \in \{1,3,5\}} \tilde{a}_{i,l}^{m,r} \tag{11}$$

Then we follow the rest of the steps in Section 3.2 and Section 3.3 to obtain classification results $\tilde{\mathbf{C}}$ and segmentation results $\tilde{\mathcal{A}}$. We show the MuSc+ results in Tab. 16. We observe that when the number of reference images is small, the classification result of MuSc+ is much lower than that of MuSc, while the segmentation result is close to that of MuSc. With the increasing number of reference images, the classification and segmentation results are also improving.

## A.7 DETAILED QUANTITATIVE RESULTS

In this section, we report the detailed results of our MuSc for each category on the MVTec AD and VisA datasets, which are presented in Tab. 17 and Tab. 18.

Table 17: Quantitative results on the **MVTec AD** dataset. All metrics are in %.

| class | AUROC-cls | F1-max-cls | AP-cls | AUROC-segm | F1-max-segm | AP-segm | PRO-segm |
|---|---|---|---|---|---|---|---|
| bottle | 99.9 | 99.2 | 100.0 | 98.6 | 79.6 | 83.2 | 96.2 |
| cable | 99.0 | 97.3 | 99.4 | 96.3 | 62.3 | 59.0 | 90.8 |
| capsule | 96.7 | 95.3 | 99.3 | 98.9 | 49.5 | 48.1 | 95.5 |
| carpet | 99.9 | 99.4 | 100.0 | 99.5 | 73.4 | 75.6 | 97.6 |
| grid | 98.7 | 96.5 | 99.5 | 98.4 | 44.6 | 39.4 | 94.7 |
| hazelnut | 99.6 | 98.6 | 99.8 | 99.4 | 74.1 | 73.8 | 92.7 |
| leather | 100.0 | 100.0 | 100.0 | 99.7 | 63.7 | 65.4 | 98.7 |
| metal_nut | 96.3 | 97.4 | 99.1 | 86.0 | 46.3 | 47.5 | 89.5 |
| pill | 96.4 | 96.2 | 99.3 | 97.6 | 66.5 | 68.0 | 98.0 |
| screw | 83.5 | 89.4 | 91.3 | 98.9 | 42.9 | 37.6 | 94.8 |
| tile | 100.0 | 100.0 | 100.0 | 98.1 | 75.3 | 78.9 | 94.9 |
| toothbrush | 100.0 | 100.0 | 100.0 | 99.5 | 70.3 | 67.5 | 95.8 |
| transistor | 99.1 | 94.7 | 98.8 | 92.0 | 59.6 | 59.0 | 78.3 |
| wood | 98.5 | 98.3 | 99.5 | 97.4 | 69.0 | 75.5 | 94.8 |
| zipper | 99.9 | 99.6 | 100.0 | 98.4 | 62.4 | 61.5 | 94.7 |
| Mean | 97.8 | 97.5 | 99.1 | 97.3 | 62.6 | 62.7 | 93.8 |

Table 18: Quantitative results on the **VisA** dataset. All metrics are in %.

| class | AUROC-cls | F1-max-cls | AP-cls | AUROC-segm | F1-max-segm | AP-segm | PRO-segm |
|---|---|---|---|---|---|---|---|
| candle | 96.2 | 91.3 | 96.2 | 99.4 | 39.4 | 28.7 | 97.8 |
| capsules | 88.8 | 86.1 | 93.9 | 98.8 | 50.5 | 44.0 | 89.3 |
| cashew | 98.6 | 96.0 | 99.3 | 99.3 | 74.5 | 77.0 | 94.3 |
| chewinggum | 98.3 | 96.0 | 99.2 | 99.5 | 59.7 | 59.2 | 88.5 |
| fryum | 99.0 | 98.0 | 99.6 | 97.8 | 57.1 | 49.5 | 94.5 |
| macaroni1 | 89.7 | 83.9 | 89.0 | 99.5 | 22.6 | 15.3 | 96.5 |
| macaroni2 | 69.9 | 70.5 | 68.9 | 97.2 | 12.4 | 4.5 | 88.8 |
| pcb1 | 89.8 | 85.6 | 89.9 | 99.5 | 80.3 | 88.2 | 92.9 |
| pcb2 | 93.4 | 88.6 | 94.3 | 97.6 | 34.8 | 22.1 | 86.6 |
| pcb3 | 93.8 | 86.8 | 93.9 | 98.2 | 41.3 | 41.6 | 92.8 |
| pcb4 | 98.4 | 93.8 | 98.3 | 98.7 | 46.1 | 44.6 | 92.5 |
| pipe_fryum | 98.4 | 97.0 | 99.2 | 99.4 | 67.4 | 67.2 | 97.5 |
| Mean | 92.8 | 89.5 | 93.5 | 98.8 | 48.8 | 45.1 | 92.7 |

## A.8 DETAILED QUALITATIVE RESULTS

In this section, we provide further qualitative results of every category on the MVTec AD and VisA datasets. We report the segmentation results of our MuSc for varying sizes and types of anomalies

on each category of both two datasets. The VisA results are from Fig. 11 to Fig. 13 and the MVTec AD results are from Fig. 14 to Fig. 16.

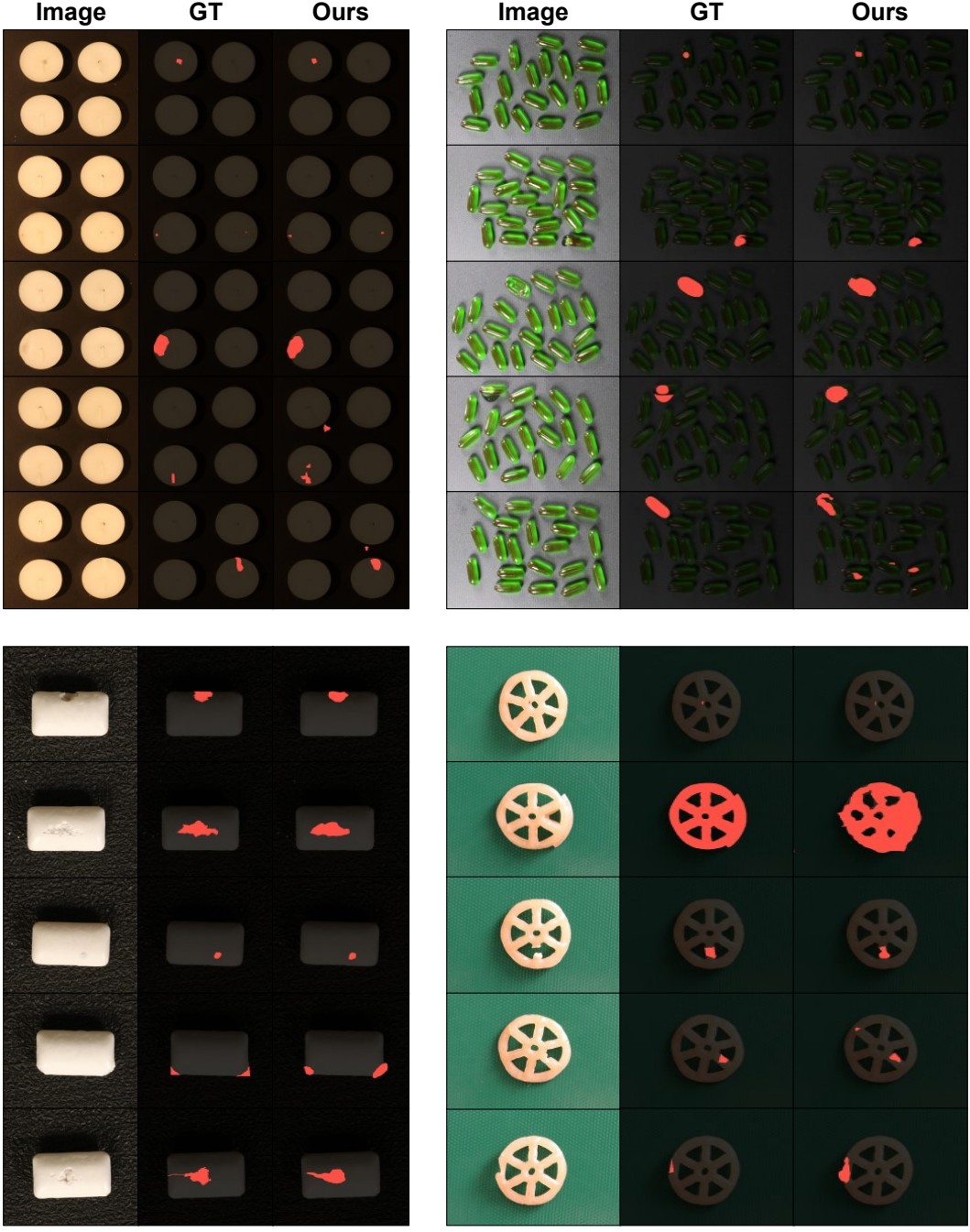

Figure 11: Qualitative results of our anomaly segmentation results on the **VisA** benchmark. In the first row, from left to right are the results for *candle* and *capsules* categories. In the second row, from left to right are the results for *chewinggum* and *fryum* categories.

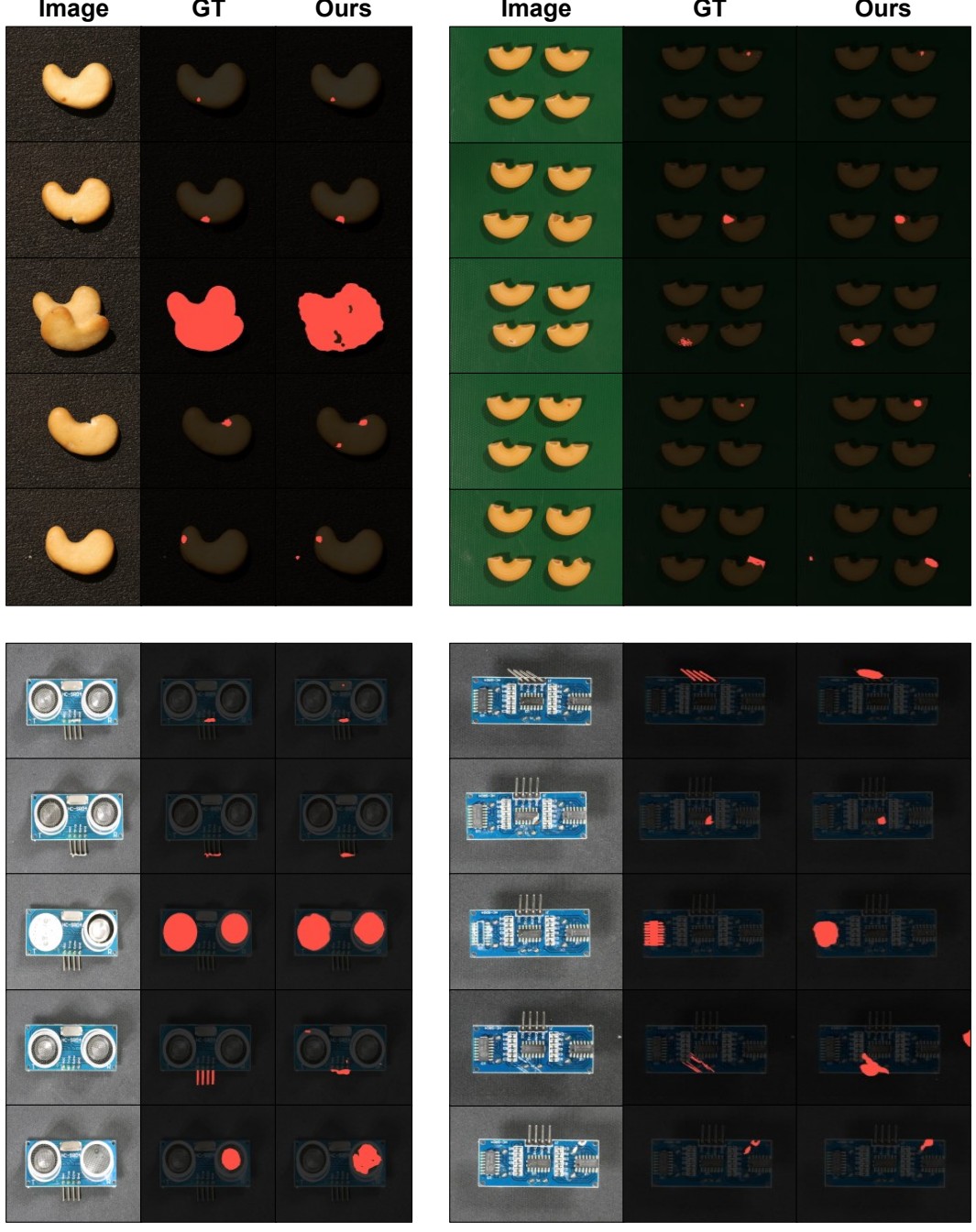

Figure 12: Qualitative results of our anomaly segmentation results on the **VisA** benchmark. In the first row, from left to right are the results for *cashew* and *macaroni1* categories. In the second row, from left to right are the results for *pcb1* and *pcb2* categories.

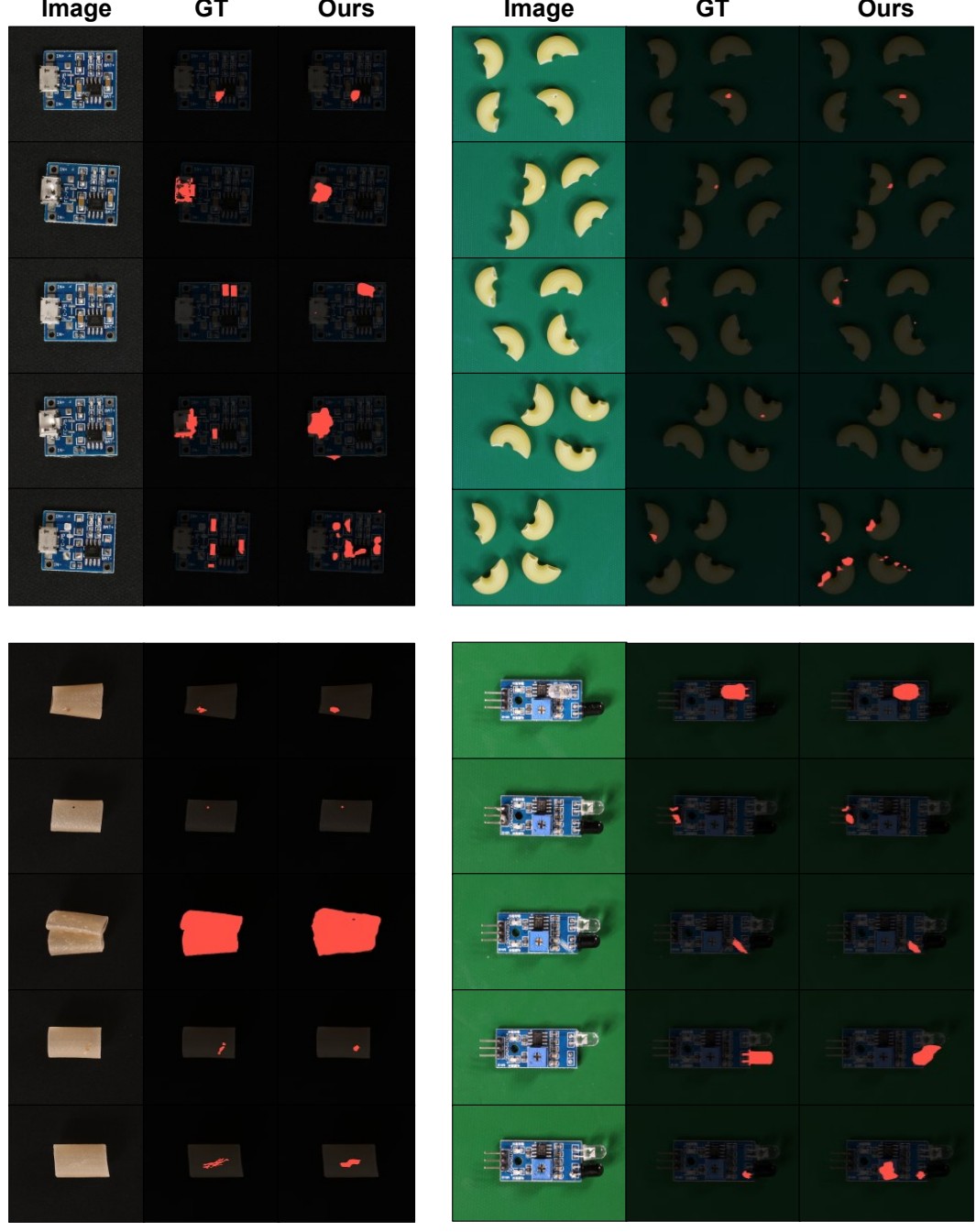

Figure 13: Qualitative results of our anomaly segmentation results on the **VisA** benchmark. In the first row, from left to right are the results for *pcb4* and *macaroni2* categories. In the second row, from left to right are the results for *pipe_fryum* and *pcb3* categories.

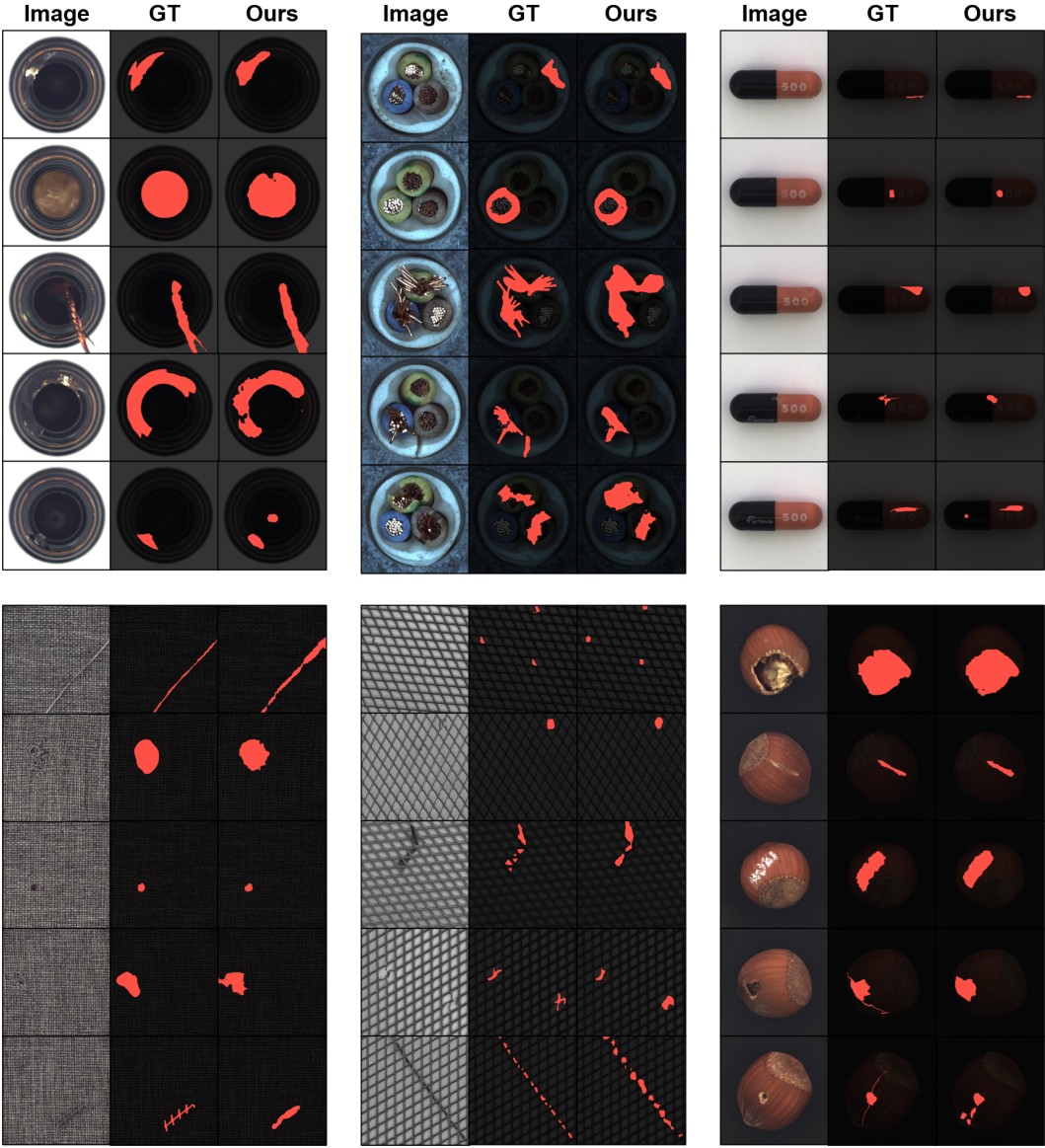

Figure 14: Qualitative results of our anomaly segmentation results on the **MVTec AD** benchmark. In the first row, from left to right are the results for *bottle*, *cable*, and *capsule* categories. In the second row, from left to right are the results for *carpet*, *grid*, and *hazelnut* categories.

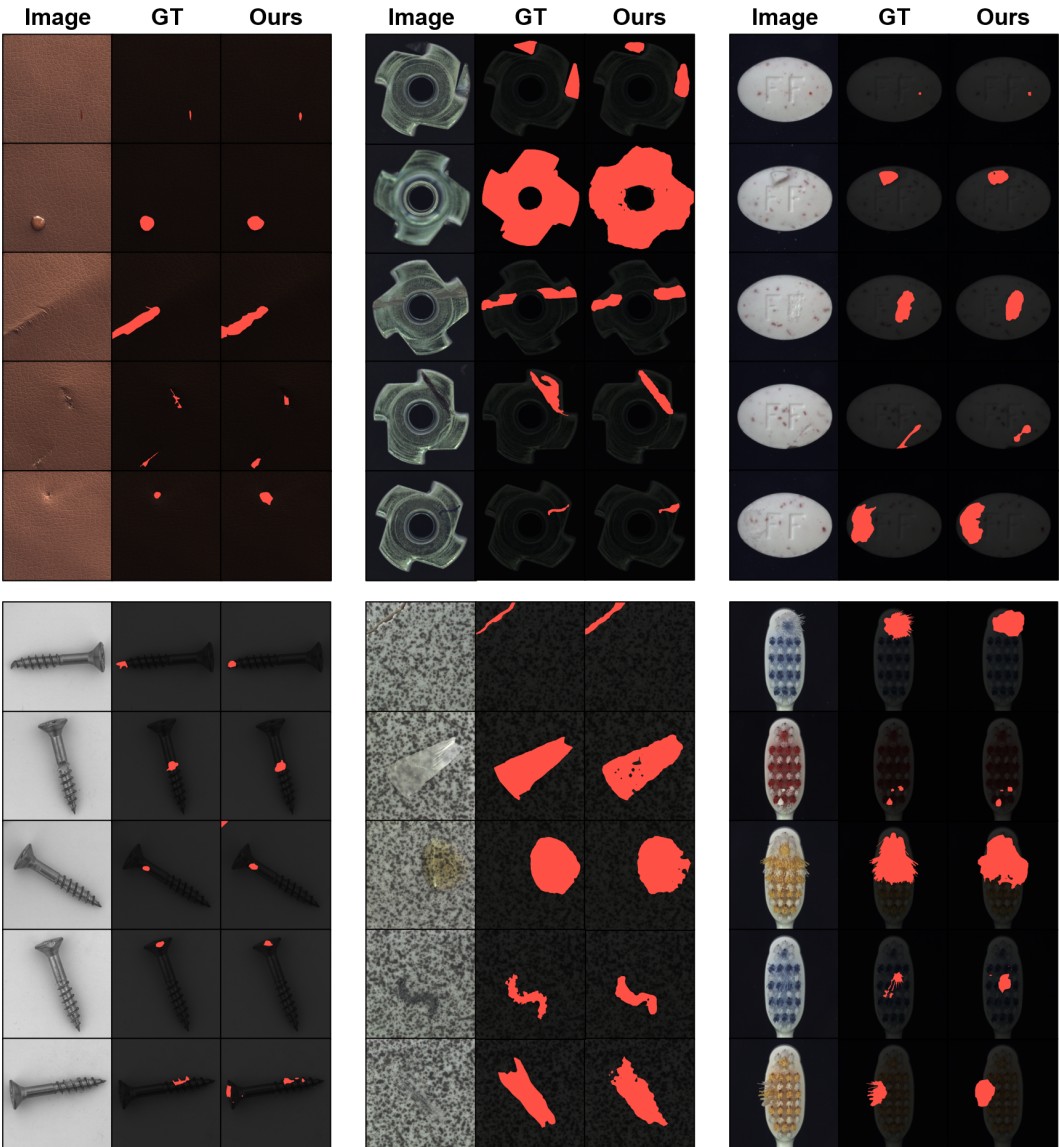

Figure 15: Qualitative results of our anomaly segmentation results on the **MVTec AD** benchmark. In the first row, from left to right are the results for *leather*, *metal_nut*, and *pill* categories. In the second row, from left to right are the results for *screw*, *tile*, and *toothbrush* categories.

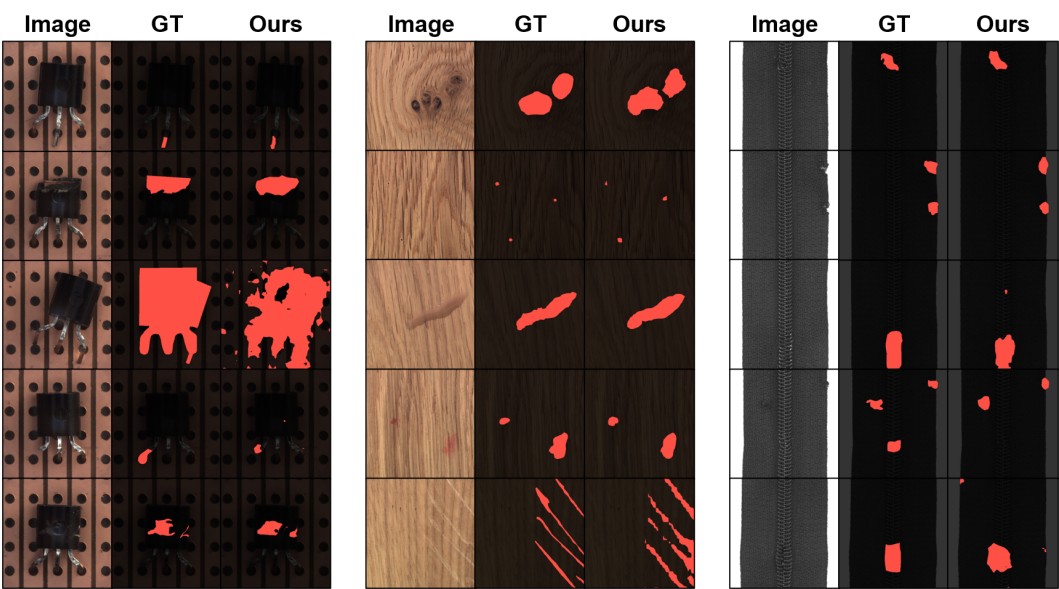

Figure 16: Qualitative results of our anomaly segmentation results on the **MVTec AD** benchmark. From left to right are the results for *transistor*, *wood*, and *zipper* categories.

