# OpenReview forum: "MuSc: Zero-Shot Industrial Anomaly Classification and Segmentation with Mutual Scoring of the Unlabeled Images"
_ICLR.cc/2024/Conference — ICLR 2024 poster_

### Official Review · Reviewer_H5pV · 2023-10-30

**Soundness:** 4 excellent
**Presentation:** 4 excellent
**Contribution:** 3 good
**Rating:** 6
**Confidence:** 5

**Summary:**

This paper targets industrial zero-shot anomaly detection. Leveraging mutual scoring on unlabeled data, the proposed method achieves SOTA performance on several well-known industrial anomaly detection benchmarks.

**Strengths:**

1.	The paper is well written, with clear architecture.
2.	The motivation is insightful, with clear ablation studies to show its effectiveness. The motivation and the proposed method are well-matched.

**Weaknesses:**

1.	This paper employs a methodology that utilizes unlabeled test images to collectively measure anomaly scores, differing somewhat from the traditional zero-shot recognition setting. In the conventional approach, each image is independently evaluated, such as WinCLIP. This difference in evaluation methodologies can result in a somewhat unfair comparison.
2.	The underlying concept is reminiscent of [a], which presumes homogeneous input texture and identifies image regions that disrupt this homogeneity as anomalies. In other words, if a patch significantly differs from its neighboring areas, it's deemed an anomaly.
3.	The approach of jointly measuring anomaly scores across entire datasets aligns closely with [b]. While not mandatory, it would be beneficial for the authors to discuss or draw comparisons with [b].

[a] Aota et al. "Zero-shot versus Many-shot: Unsupervised Texture Anomaly Detection." In WACV 2023.

[b] Li et al. "Zero-Shot Batch-Level Anomaly Detection." arXiv, February 2023.

**Questions:**

See the weakness.

---

> ### Author Response · Authors · 2023-11-22
> **Response to Reviewer H5pV**
>
> Thanks for the insightful comments.
>
> **Q1: This paper employs a methodology that utilizes unlabeled test images to collectively measure anomaly scores, differing somewhat from the traditional zero-shot recognition setting. This difference in evaluation methodologies can result in a somewhat unfair comparison.**
>
> A1: We agree that our approach is different from several existing approaches, e.g., winCLIP, in using the unlabeled test images, which may lead to a potentially unfair comparison. Therefore we have added the comparison to ACR [b]\[c] in Tab.1.  Similar to our approach, ACR assumes access to a set of unlabeled test images together during inference, the experimental results demonstrate that MuSc outperforms ACR by a large margin. We hope such a comparison is more fair.
>
> **Q2: The underlying concept is reminiscent of [a], which presumes homogeneous input texture and identifies image regions that disrupt this homogeneity as anomalies.**
>
> A2: Thanks for your valuable suggestion! We have carefully studied [a] and added the discussion in the Related Work section. [a] is based on the assumption that the input texture is homogeneous, and the image regions that break the homogeneity are detected as anomalies. [a] explores the relationship between the patches inside one test texture image, while our method leverages the relationship in a set of unlabeled images.
>
> **Q3: The approach of jointly measuring anomaly scores across entire datasets aligns closely with [b]. While not mandatory, it would be beneficial for the authors to discuss or draw comparisons with [b].**
>
> A3: Thanks for your valuable suggestion! We have carefully studied ACR [b]\[c] and added the comparison with it in Tab. 1 and the discussion in the Related Works section. We found that ACR is an excellent zero-shot AD approach, we agree that our approach aligns closely with ACR, since both of us jointly measure the anomaly scores across the entire dataset or a set of unlabeled test images, while we use the unlabeled images differently, another difference is that ACR contains a training process, while our approach does not need any training or prompt.
>
> We give the classification and segmentation performance of our method and [b]\[c] on the MVTec AD dataset in the following table.
>
> |   Method    | AUROC-cls | AUROC-segm | PRO-segm |
> | :---------: | :-------: | :--------: | :------: |
> |   ACR [b]   |   85.8    |    92.5    |   72.7   |
> | MuSc (ours) |   97.8    |    97.3    |   93.8   |
>
> > [a] Aota et al. "Zero-shot versus Many-shot: Unsupervised Texture Anomaly Detection." In WACV 2023.
> >
> > [b] Li et al. "Zero-Shot Batch-Level Anomaly Detection." arXiv, February 2023.
> >
> > [c] Li et al. "Zero-Shot Anomaly Detection via Batch Normalization." In NeurIPS 2023.
> >

---

### Official Review · Reviewer_dCfU · 2023-10-30

**Soundness:** 3 good
**Presentation:** 3 good
**Contribution:** 2 fair
**Rating:** 5
**Confidence:** 4

**Summary:**

This paper addresses zero-shot anomaly classification (AC) and segmentation (AS) with following contributions:
1) Using unlabeled test images for AC/AS.
2) A new mutual scoring mechanism for identification of abnormal patches.
3) SOTA performance, significantly outperforming existing zero-shot methods

**Strengths:**

The claimed contributions summarized above.

**Weaknesses:**

1) This paper makes the assumption that access to the entire test dataset is available. This allows for a direct application of the proposed mutual scoring mechanism. However, for zero-shot settings, such an assumption is too restrictive.  Access to test data is mostly very limited in real-world scenarios, especially for zero-shot.

2) The method heavily relies on many heuristic design choices. It consists of a three-step pipeline which depends on many hyperparameters for feature representation, mutual scoring estimation, and classification rescoring. The paper poorly presents sensitivity of performance to optimizing all these parameters.

3) Overall technical novelty seems incremental, since the method incorporates well-established multiscale features (Sec 3.1), norm-2 distance (Sec 3.2), and clustering (Sec 3.3).

4) Since the method lacks a training phase and directly generates results based on estimating the test-set statistics, inference is likely to be much slower than in previous approaches. I was not able to find a report/comparison of the inference times.

**Questions:**

1) Could your approach handle a setting when a test dataset lacks ground truth? What is performance when a test dataset lacks ground truth for optimizing the proposed heuristic procedure?

2) One of the claimed contributions, the mutual scoring mechanism, appears to depend significantly on the relative positions of feature patches. This limits the application of this approach in real settings where orientations and scales are not consistent. What is performance when test images exhibit inconsistent orientations or scales?

3) Could your method be extended to (or readily address) the few-shot setting?

---

> ### Author Response · Authors · 2023-11-22
> **Response to Reviewer dCfU (Part 1)**
>
> Thanks for your insightful comments and thoughtful questions.
>
> **Q1: This paper makes the assumption that access to the entire test dataset is available. For zero-shot settings, such an assumption is too restrictive. Access to test data is mostly very limited in real-world scenarios, especially for zero-shot.**
>
> A1: Thanks for your comment! As mentioned by Reviewer H5pV, our approach aligns closely with ACR [a]\[b], an excellent zero-shot AD approach published in NeurIPS2023, both of us jointly measure the anomaly scores across either the entire dataset or a sub-set of unlabeled test images. Therefore, we hope the comparison with ACR makes our approach more fair.
>
> **Q2: The method heavily relies on many heuristic design choices. It consists of a three-step pipeline which depends on many hyperparameters. The paper poorly presents the sensitivity of performance to optimizing all these parameters.**
>
> A2: Our method contains three steps, and each step has only one hyperparameter, i.e., the aggregation degree $r$ in LNAMD, Interval Average (IA) on the minimum $X\%$, and window mask size in RsCIN.
>
> The setting of aggregation degree $r$ indeed impacts the result as it is related to the size of anomalies. As shown in Tab. 3, in multiple settings, the model's classification AUROC on the MVTec AD dataset is reduced by up to 3.6%, and segmentation AUROC is reduced by up to 3%. On the VisA dataset, the use of $r$={5} results in a 10.4% reduction in classification results compared to $r$={1,3,5}, due to more small defects on VisA, making it unsuitable to use only the large aggregation degree. For better practicality, we use the same setting {1,3,5} in all the datasets to obtain a good segmentation of anomalies with various sizes.
>
> The $X\%$ in Interval Average is an insensitive hyperparameter. As shown in Fig. 7, we compare the models using different percentage ranges of the Interval Average in the MSM module. The change of this parameter has a maximum effect of 3% on the image-AUROC and 1% on the pixel-AUROC.
>
> The window mask size in RsCIN is also an insensitive hyperparameter, which is proved by the experiments in Fig. 8. Observably, on the MVTec AD data set, F1-max-score fluctuates between 97.1 and 97.5, with the amplitude not exceeding 0.4. On the VisA data set, F1-max-score floats between 87.8 and 89.5, and the amplitude does not exceed 1.7. Furthermore, we also verify the sensitivity of the RsCIN module to different methods, as shown in Tab. 11 of the appendix. All methods maintain the same parameter settings of the RsCIN module as ours. Almost all methods achieve performance improvements after applying the RsCIN module, except DRAEM only. This experiment proves that the window mask size in RsCIN is insensitive for different methods.
>
> In summary, the aggregation degree $r$ is a sensitive parameter, while the other two hyperparameters are insensitive.
>
> In addition, we show the performance of our method on the BTAD dataset, and the parameter settings are consistent with those in our experiments. Specific settings include aggregation degree $r$ as {1, 3, 5},  $X$ as 30, and window mask size as {2, 3}. The following table shows that the classification and segmentation performance of our method on the BTAD dataset still exceeds that of zero/few-shot methods and even some full-shot methods. Such an experimental result shows that although the performance of MuSc is sensitive to the setting of aggregation degree $r$, an appropriate choice of $r$ can achieve good results in different datasets simultaneously, which proves its universality.
>
> |   Method    |  Setting  | Image-AUROC | Pixel-AUROC |
> | :---------: | :-------: | :---------: | :---------: |
> |   VT-ADL    | full-shot |    83.7     |    90.0     |
> |   P-SVDD    | full-shot |    83.3     |    92.1     |
> |    SPADE    | full-shot |    87.6     |    96.9     |
> |    PaDiM    | full-shot |    93.7     |    97.3     |
> | PyramidFlow | full-shot |    95.8     |    97.7     |
> |  PatchCore  |  4-shot   |    91.4     |    96.3     |
> |    RegAD    |  4-shot   |    91.0     |    97.5     |
> |  APRIL-GAN  |  4-shot   |    91.3     |    92.4     |
> |  APRIL-GAN  |  0-shot   |    72.3     |    89.6     |
> | MuSc(ours)  |  0-shot   |    94.8     |    97.3     |

---

> > ### Author Response · Authors · 2023-11-22
> > **Response to Reviewer dCfU (Part 2)**
> >
> > **Q3: Overall technical novelty seems incremental, since the method incorporates well-established multiscale features (Sec 3.1), norm-2 distance (Sec 3.2), and clustering (Sec 3.3).**
> >
> > A3: Thanks for your comment! We clarify our technique novelty as follows.
> >
> > Firstly, we find that the patch-level features extracted by ViT have the limitation of detecting industrial anomalies of varying sizes. Swin transformer has large patches in some stages leading to the patch-level features extracted by it not working well. The proposed LNAMD module addresses this issue in ViT, as far as we know, such a design is a new technique to optimize the patch-level features of ViT in industrial vision.
> >
> > Secondly, in MSM, the mutual scoring operation in Eq. (1) is inspired by the observation of normal and abnormal patches. It uses the minimum similarity between the patch $m$ of image $I_i$ and all the patches of image $I_j$ to score the patch $m$ of image $I_i$. This is a new technique to score the patch by unlabeled images. The interval average operation in Eq. (2) is based on our observation of the false positives, so we designed this operation to alleviate these in scoring. To the best of our knowledge, the Mutual Scoring Mechanism is a new mechanism for the zero-shot AC/AS community.
> >
> > Thirdly, in RsCIN, we observe that due to the high smoothness of $\textbf{C}$, the traditional manifold learning approach may be influenced by a large number of images. Therefore, we propose the Multi-window Mask Operation (MMO) to constrain the image number, which makes each image only influenced by a small number of neighborhood images. RsCIN was first proposed in the field of anomaly classification and segmentation. Meanwhile, we experimentally proved that RsCIN can be further employed to improve the AC accuracy of existing approaches in Appendix A.2.4.
> >
> > **Q4: Since the method lacks a training phase and directly generates results based on estimating the test-set statistics, inference is likely to be much slower than in previous approaches. I was not able to find a report/comparison of the inference times.**
> >
> > A4: Thanks for your insightful comment. Driven by your suggestion, we have examined the MuSc's code implementation thoroughly and found two issues that slowed down the model and increased the number of parameters. For now, both issues have been solved. After the paper is accepted, we will release the source code to prove that both issues are indeed addressed and the parameter amount and speed are optimized. We describe these two issues and the corresponding countermeasures in the response to Q1 of reviewer NLA8.
> >
> > After modifying the implementation of LNAMD, we have conducted a **comprehensive efficiency comparison**, which is listed in the following table. Since there is no official source code of WinCLIP, its GPU cost is unknown (marked by “-”). In addition, the author of APRIL-GAN does not provide the pre-trained model using ViT-B-16-plus-240 (our backbone), so its AUROC and AUPRO are unknown (marked by “-”).
> >
> > |      Method       |     Backbone      | Training | Inference time (ms) | GPU (MB) | Image-AUROC | AUPRO |
> > | :---------------: | :---------------: | :------: | :-----------------: | :------: | :---------: | :---: |
> > |   RegAD(4-shot)   |     ResNet18      |   yes    |        309.4        |   8920   |    89.1     | 88.0  |
> > | APRIL-GAN(0-shot) |   ViT-L-14-336    |   yes    |        100.2        |   4996   |    86.1     | 44.0  |
> > |    MuSc($s=1$)    |   ViT-L-14-336    |    no    |        998.8        |   7168   |    97.8     | 93.8  |
> > |    MuSc($s=2$)    |   ViT-L-14-336    |    no    |        605.8        |   5666   |    97.1     | 93.8  |
> > |    MuSc($s=3$)    |   ViT-L-14-336    |    no    |        513.5        |   5026   |    96.7     | 93.7  |
> > | APRIL-GAN(0-shot) | ViT-B-16-plus-240 |   yes    |        64.7         |   3226   |      -      |   -   |
> > |  WinCLIP(0-shot)  | ViT-B-16-plus-240 |    no    |         389         |    -     |    91.8     | 64.6  |
> > |    MuSc($s=1$)    | ViT-B-16-plus-240 |    no    |        455.3        |   3002   |    95.4     | 91.9  |
> > |    MuSc($s=2$)    | ViT-B-16-plus-240 |    no    |        285.7        |   2740   |    95.4     | 91.5  |
> > |    MuSc($s=3$)    | ViT-B-16-plus-240 |    no    |        238.3        |   2648   |    95.2     | 91.4  |
> >
> > In terms of inference time, when using the same backbone as WinCLIP and setting $s$ to 3, our inference time is 238.3ms per image, which is 150.7ms shorter than WinCLIP, and 71.1ms shorter than RegAD. Additionally, APRIL-GAN has the shortest inference time (64.7ms) among these methods. In terms of memory cost, when using the same backbone as WinCLIP, our GPU memory cost is 3002MB, significantly lower than RegAD (8920MB) and 224MB less than APRIL-GAN (3226MB). The detailed comparisons of inference time and memory cost have been added to Appendix A.4.
> >
> > We will continue to speed up MuSc and reduce its memory cost in our future works.

---

> ### Author Response · Authors · 2023-11-22
> **Response to Reviewer dCfU (Part 3)**
>
> **Q5: Could your approach handle a setting when a test dataset lacks ground truth? What is performance when a test dataset lacks ground truth for optimizing the proposed heuristic procedure?**
>
> A5: Thanks for your comment. As we mentioned in the response to Q2 of reviewer dCfU, our approach contains three hyperparameters. Our performance is sensitive to the aggregation degree $r$ but insensitive to the other two hyperparameters. In addition, the experimental results on the new BTAD dataset demonstrate that the fixed hyperparameter setting could achieve good experimental results in different datasets. Therefore, we do not need to use the ground truth to optimize our approach. It is clarified that all the steps of our approach do not use the ground truth actually.
>
> **Q6: One of the claimed contributions, the mutual scoring mechanism, appears to depend significantly on the relative positions of feature patches. This limits the application of this approach in real settings where orientations and scales are not consistent. What is performance when test images exhibit inconsistent orientations or scales?**
>
> A6: Thanks for your comment! In the mutual scoring mechanism, we leverage each image in {$D_{test}$ \\ $I_i$} to assign an anomaly score for each aggregated patch token of a test image $I_i$ in stage one as follows，
>
> $$
> a_{i,l}^{m,r}(I_j) = \\min_{n} \\Vert \\hat{p}_{i,l}^{m,r}-\\hat{p}^{n,r}\_{j,l} \\Vert_2
> $$
>
> where $m$ and $n$ are the index of the image patches, $r$ is the aggregation degree, $i$ and $j$ are the index of the unlabeled images, and $l$ indicates the stage $l$ of ViT. We clarify that $m$ is irrelevant to the position of $n$,  since Eq.1 first computes the distance between the  $m$-th patch (in image $I_i$) and all the patches in image $I_j$, and then performs min operation to select the most similar image patch from image  $I_j$. Accordingly, the anomaly score only depends on the aggregated patch token and is irrelevant to the relative positions of feature patches.
>
> In addition, according to your insightful comment, we select the categories with inconsistent orientations or scales from MVTec AD and VisA datasets for the additional experimental evaluation, which is given in Tab. 12 in the paper and also given in the following table. We compare our method with WinCLIP and APRIL-GAN and show the image-AUROC (AC)/pixel-AUROC (AS) in this table. We observe that our approach and WinCLIP are both influenced by inconsistent orientations or scales. The reason is that both of us use the fixed pre-trained vision transformer as the feature extractor. As introduced in [c], during the pre-training process, the vision transformer only performs minor data augmentation on orientations and scales, and hence both of us cannot well address the inconsistent orientations or scales. APRIL-GAN also achieves a decreased AC score, while it obtains a better AS score. We guess that it is due to its additional training set for optimizing AS. In addition, we should emphasize that our approach still outperforms WinCLIP and APRIL-GAN in inconsistent orientations or scale cases.
>
> In our future works, we plan to design rotation and scale invariance features to alleviate such a limitation.
>
> | Category  |    WinCLIP    |   APRIL-GAN   |  MuSc(ours)   |
> | :-------: | :-----------: | :-----------: | :-----------: |
> |           |     AC/AS     |     AC/AS     |     AC/AS     |
> |   screw   |   83.3/89.6   |   84.9/97.8   |   83.5/98.9   |
> | hazelnut  |   93.9/94.3   |   89.6/96.1   |   99.6/99.4   |
> | metal_nut |   97.1/61.0   |   68.4/65.4   |   96.3/86.0   |
> | capsules  |   85.0/81.6   |   61.2/97.5   |   88.8/98.8   |
> | macaroni2 |   63.7/59.3   |   64.6/97.8   |   69.9/97.2   |
> | **mean**  | **84.6/77.2** | **73.7/90.9** | **87.6/96.1** |
> | mean-ALL  |   91.8/85.1   |   86.1/87.6   |   97.8/97.3   |

---

> > ### Author Response · Authors · 2023-11-22
> > **Response to Reviewer dCfU (Part 4)**
> >
> > **Q7: Could your method be extended to (or readily address) the few-shot setting?**
> >
> > A7: Thanks for your valuable comment. We provided two simple few-shot setting extensions based on MuSc in Appendix A.6. In the first extension, we added a few normal reference images directly to the test images that assign scores to image $I_i$ in the MSM module. In the second extension method, we designed additional scoring strategies for these normal reference images. In detail, we modified the original Eq. 2 to replace the interval average operation with the minimum operation. Since our zero-shot method uses rich implicit normal prior information in the unlabeled dataset, adding a few reference images does not greatly improve the results. The results of the two extended few-shot settings are in Tab. 14 and Tab. 15. In addition, as shown in Tab. 1, our zero-shot approach outperforms several few-shot approaches.
> >
> > > [a] Li et al. "Zero-Shot Batch-Level Anomaly Detection." arXiv, February 2023.
> > >
> > > [b] Li et al. "Zero-Shot Anomaly Detection via Batch Normalization." In NeurIPS 2023.
> > >
> > > [c] Radford et al "Learning Transferable Visual Models From Natural Language Supervision" In ICML 2021.

---

### Official Review · Reviewer_idjR · 2023-11-01

**Soundness:** 3 good
**Presentation:** 3 good
**Contribution:** 3 good
**Rating:** 6
**Confidence:** 3

**Summary:**

The paper introduces MuSc, a novel zero-shot framework for industrial anomaly classification and segmentation. It utilizes cues from unlabeled test images and combines local patch tokens with a mutual scoring mechanism. The method notably outperforms existing zero-shot approaches and rivals many few-shot and full-shot methods.

**Strengths:**

1. The MuSc framework introduces a novel approach to zero-shot anomaly classification and segmentation, particularly in the industrial domain. The method of leveraging implicit cues from unlabeled test images for anomaly detection is an innovative concept.

2. The empirical results demonstrate a substantial improvement over existing zero-shot approaches and competitiveness with few-shot and full-shot methods.

3. The approach holds significant potential for industrial applications, where anomaly detection is crucial but training data is often scarce or expensive to obtain.

**Weaknesses:**

In Table 3, the ablation study of LNAMD with different aggregation degrees \(r\) raises a question regarding the effectiveness of combining aggregation degrees. Specifically, it's unclear why the combination of \({3, 5}\) performs worse in the anomaly classification (AC) task than using \({3}\) alone. This observation seems to contradict the paper's claim that using all aggregated patch tokens with different degrees is beneficial for detecting anomalies of various sizes. Based on this claim, one would expect the combination of \({3, 5}\) to outperform either \({3}\) or \({5}\) individually. This inconsistency warrants further clarification or investigation to reconcile the results with the stated claims.

**Questions:**

See the weaknesses.

---

> ### Author Response · Authors · 2023-11-22
> **Response to Reviewer idjR**
>
> Thanks for your valuable comments.
>
> **Q1: It's unclear why the combination of ({3, 5}) performs worse in the anomaly classification (AC) task than using ({3}) alone.**
>
> A1: Thanks for your comment. We have explained this phenomenon deeply in the revised Appendix A.2.2. As we written in Section 3.1, aggregation degree ({5}) is suitable for detecting large defects, but it in turn smoothes small defects to a lower score. We show a classic example in Fig. 9 of the revised manuscript. When directly summing the anomaly scores of the two aggregation degrees ({3,5}), the score within the small anomaly region is pulled down by the low score generated by the aggregation degree ({5}), which results in false negative. Consequently, the combination of ({3,5}) performs worse in the anomaly classification (AC) task than using ({3}) alone.
>
> Due to the above reasons, we use the aggregation degrees ({1,3,5}) to take into account various-size anomalies in real industrial scenarios. As displayed in Fig 9, when we add a large anomaly score with aggregation degree ({1}), the anomaly score within the black rectangle in (e) increases and false negatives reduce.

---

### Official Review · Reviewer_52S5 · 2023-11-01

**Soundness:** 1 poor
**Presentation:** 1 poor
**Contribution:** 1 poor
**Rating:** 3
**Confidence:** 3

**Summary:**

The paper proposes an anomaly detection and segmentation method utilizing unlabeled images from a test set. It is assumed that a set of images with some form of anomaly is available for inference and it is claimed to be zero shot.
The method starts with computing a representation for every image patch through aggregating (pooling) token features generated by ViT at different layers. An anomaly score for each patch is then computed by comparing the aggregated patch token with that from each image in the *test* set. Further heuristic tricks are applied to refine this score and the final pixel level anomaly score is given by the max of the refined score.
The method then produces an image level classification by defining a weighted graph over *test* set images where weights are defined by the class token generated by ViT. The image level classification score is the determined by by some graph operations that were not explained well. The method was tested on public datasets and compared with solid baselines.

**Strengths:**

I am afraid I was not able to find a notable strength of this paper.

**Weaknesses:**

1. Technical soundness: The method relies heavily on availability of a set of images in order to compute the anomaly scores. I dont think this emulates the practical scenario of anomaly detection in industry. The more realistic scenario is a method is required to classify and segment the anomaly given one image. Using test set images to produce the output on the same set of images also does not conform with the scientific procedure. I dont think this is the definition of a zero shot method either. The setting sounds completely unreasonable to me.

2. Contribution: The algorithm appears to be a set of heuristic tricks applied in a sequence. There is not a solid technique that is novel, elegant, theoretically justified and of broad interest.

3. Clarity: None of the techniques were adequately explained as to why they are being performed and why it makes sense (intuitively or conceptually) to apply them. Just stating the process does not qualify as a good scientific exposition.

**Questions:**

...

---

> ### Author Response · Authors · 2023-11-22
> **Response to Reviewer 52S5 (Part 1)**
>
> Thanks for your comments, we want to first give the following clarifications.
>
> > “It is assumed that a set of images with some form of anomaly is available for inference”.
> >
>
> Here, it needs to be clarified that our approach leverages a set of unlabeled images for anomaly AC/AS but does not assume that the anomaly has the same form.
>
> **Q1: The method relies heavily on availability of a set of images in order to compute the anomaly scores. I dont think this emulates the practical scenario of anomaly detection in industry.**
>
> A1: Thanks for your comment. Since we have long-term collaboration with several well-known companies, our approach is designed according to the practical industry requirements.  Let us take a famous anonymous company as an example. In general, a production line (AMOLED/LCD) has a daily output of 1500 pieces. Typically, a production line is equipped with at least 3 parallel quality inspection equipment, and each furnished with at least 2 cameras. The image resolution of each camera exceeds 4000×5000, and a typical solution is to crop a series of 512×512 images from high-resolution images by sliding windows with 10% overlap. The above production line needs to inspect nearly **a million product images per day**. As far as we know, such a production quality inspection requirement exists in a large number of companies. This example illustrates that unlabeled images are abundant in real industrial scenarios and easy to collect.
>
> In addition, it needs to be clarified that MuSc did not rely heavily on a large number of unlabeled images, which can be demonstrated by Tab. 7 of the paper. MuSc obtains decent AUROCs under the test sets of different amounts, even when the test set only has 14-56 images. Referring to the example above, we are confident that such a number of images can be easily achieved in practical scenarios.
>
> **Q2：The more realistic scenario is a method is required to classify and segment the anomaly given one image. Using test set images to produce the output on the same set of images also does not conform with the scientific procedure. I dont think this is the definition of a zero shot method either.**
>
> A2: Thanks for your comment. We agree that it is a widely-used practice to classify and segment anomalies in one image. However, this does not negate the scientific validity of using a set of test images to produce the output on the same set of images. As mentioned by Reviewer H5pV, our approach aligns closely with ACR [b]\[c], an excellent zero-shot AD approach published in NeurIPS2023. Both of us jointly measure the anomaly scores across either the entire test dataset or a sub-set of unlabeled test images. Therefore, we hope the comparison with ACR makes our approach more fair and dispels your doubts.
>
> **Q3: The algorithm appears to be a set of heuristic tricks applied in a sequence. There is not a solid technique that is novel, elegant, theoretically justified and of broad interest.**
>
> A3: Thanks for your comment! We clarify our technique novel as follows.
>
> Firstly, we find that the patch-level features extracted by ViT have the limitation of detecting industrial anomalies of varying sizes. Swin transformer has large patches in some stages leading to the patch-level features extracted by it not working well. The proposed LNAMD module addresses this issue in ViT, as far as we know, such a design is a new technique to optimize the patch-level features of ViT in industrial vision.
>
> Secondly, in MSM, the mutual scoring operation in Eq. (1) is inspired by the observation of normal and abnormal patches. It uses the minimum similarity between the patch $m$ of image $I_i$ and all the patches of image $I_j$ to score the patch $m$ of image $I_i$. This is a new technique to score the patch by unlabeled images. The interval average operation in Eq. (2) is based on our observation of the false positives, so we designed this operation to reduce the false positives. To the best of our knowledge, the Mutual Scoring Mechanism is a new mechanism for the zero-shot AC/AS community.
>
> Thirdly, in RsCIN, we observe that due to the high smoothness of $\textbf{C}$, the traditional manifold learning approach may be influenced by a large number of images. Therefore, we propose the Multi-window Mask Operation (MMO) to constrain the image number, which makes each image only influenced by a small number of neighborhood images. RsCIN was first proposed in the field of anomaly classification and segmentation. Meanwhile, we experimentally proved that RsCIN can be further employed to improve the AC accuracy of existing approaches in Appendix A.2.4.

---

> > ### Author Response · Authors · 2023-11-22
> > **Response to Reviewer 52S5 (Part 2)**
> >
> > **Q4: None of the techniques were adequately explained as to why they are being performed and why it makes sense (intuitively or conceptually) to apply them. Just stating the process does not qualify as a good scientific exposition.**
> >
> > A4: Thanks for your comment, we clarify the motivations as follows.
> >
> > The LNAMD module is motivated by the observation that *the large aggregation degree $r​$ is suitable for detecting large anomaly regions, and a small aggregation degree $r​$ is suitable for detecting small abnormal regions*, which are described in the second paragraph of Section 3.1.
> >
> > The MSM contains three steps, and the motivations are given as follows. The motivation of the first step is that *the normal patch in image $I_i$ can find similar patches in most other images, while the abnormal patch is difficult to find similar patches* (line 9 to line 13 in the first paragraph of Section 3.2). The second step is inspired by our observation of the anomaly scores histogram. We find that *some normal patches with appearance variations of different normal images may cause false positives*, so we take the Interval Average operation to reduce them in scoring (line 2 to line 6 in the second paragraph). The third step, which aims to *achieve the fine-grained anomaly score for each image patch*, is described in the first line of the third paragraph.
> >
> > For the motivation of RsCIN, as we described in the first paragraph of Section 3.3. We observed that *images similar to normal images have smaller classification anomaly scores, while those similar to abnormal images have larger classification anomaly scores*, which inspired us to use these similar images to optimize classification scores and design this module.
> >
> > **Q5: The image level classification score is the determined by some graph operations that were not explained well.**
> >
> > A5: In the revised manuscript, we have detailed the theory of our RsCIN module in Appendix A.1.2.

---

### Official Review · Reviewer_NLA8 · 2023-11-02

**Soundness:** 2 fair
**Presentation:** 2 fair
**Contribution:** 3 good
**Rating:** 6
**Confidence:** 5

**Summary:**

This paper presents a MuSc model for zero-shot AC/AS. The method exploits the normal and abnormal information implicit in unlabeled test images without any training or prompts, A Mutual Scoring Mechanism (MSM) is proposed to assign abnormal scores to each other using unlabeled test images, and an optimization method based on constrained image-level neighborhoods (RsCIN) for image-level anomaly classification. The method shows better performance on MVTec AD and VisA datasets.

**Strengths:**

1. The idea that normal image patches can be found in a relatively large number of similar patches in other unlabeled images, while abnormal image patches have only a small number of similar patches, is novel. The authors were able to accomplish this task simply by using a test dataset and utilizing the test images to score each other without any training or prompting.

2. The authors considered the size of the anomalies in different datasets and used patch tokens with multiple aggregation degrees to obtain high-quality anomaly scores even when using a simple distance measure.

3. The authors found that the image-level features satisfy the conditions of high-dimensional manifolds, and designed RsCIN based on manifold learning to optimize the pixel-level anomaly classification results, and experimentally verified the effectiveness of the module, proving that the proposed RsCIN module can further improve the performance of the existing methods.

**Weaknesses:**

Even without training or prompts, the method requires long inference times and high memory costs. While the authors provide a solution to increase speed and reduce memory by dividing the test set into subsets, this also reduces performance by a small margin. Are there any other approaches that could have been considered to solve the problem?

The authors should provide more detailed theories for MSM and RsCIN.

**Questions:**

See above

---

> ### Author Response · Authors · 2023-11-22
> **Response to Reviewer NLA8 (Part 1)**
>
> Thanks for your insightful comments.
>
> **Q1: Even without training or prompts, the method requires long inference times and high memory costs. While the authors provide a solution to increase speed and reduce memory by dividing the test set into subsets, this also reduces performance by a small margin. Are there any other approaches that could have been considered to solve the problem?**
>
> A1:  Driven by your suggestion, we have examed the MuSc's code implementation thoroughly and found two issues that slowed down the model and increased the number of parameters. For now, both issues have been solved. After the paper is accepted, we will release the source code to prove that both issues are indeed addressed and the parameter amount and speed are optimized. Below, we describe these two issues and the corresponding countermeasures respectively.
>
> **First issue: A time-consuming operation in LNAMD.** In the original implementation of the LNAMD, we have used the "for" loop operation to process a tensor with dimensions [1369, 1024], i.e.,  "features", resulting in a lot of time-consuming.
>
> `return [x.detach().cpu().numpy() for x in features]`
>
> Consequently, this inefficient implementation costs 95.2ms under a specific aggregation degree r, consuming more than 70% of the time in LNAMD which takes 129.9ms only. We have addressed this issue by the following code:
>
> `return features.detach().cpu().numpy()`
>
> **Second issue: Additional GPU memory usage in LNAMD.** The original implementation of the LNAMD module unexpectedly loaded the backbone network an extra time, which was neglected when we modified the code structure. For now, we have removed the extra process of loading the backbone, saving 1186MB and 620MB of memory respectively when using ViT-L-14-336 and ViT-B-16-plus-240.
>
> After modifying the code of LNAMD, we have conducted a **comprehensive efficiency comparison**, which is listed in the following table. Since there is no official source code of WinCLIP, its GPU cost is unknown (marked by “-”). In addition, the author of APRIL-GAN does not provide the pre-trained model using ViT-B-16-plus-240 (our backbone), so its AUROC and AUPRO are unknown (marked by “-”).
>
> In terms of inference time, when using the same backbone as WinCLIP and setting $s$ to 3, our inference time is 238.3ms per image, which is 150.7ms shorter than WinCLIP, and 71.1ms shorter than RegAD. Additionally, APRIL-GAN has the shortest inference time (64.7ms) among these methods. In terms of memory cost, when using the same backbone as WinCLIP, our GPU memory cost is 3002MB, significantly lower than RegAD (8920MB) and 224MB less than APRIL-GAN (3226MB). The detailed comparisons of inference time and memory cost have been added to Appendix A.4.
>
> |      Method       |     Backbone      | Training | Inference time (ms) | GPU (MB) | Image-AUROC | AUPRO |
> | :---------------: | :---------------: | :------: | :-----------------: | :------: | :---------: | :---: |
> |   RegAD(4-shot)   |     ResNet18      |   yes    |        309.4        |   8920   |    89.1     | 88.0  |
> | APRIL-GAN(0-shot) |   ViT-L-14-336    |   yes    |        100.2        |   4996   |    86.1     | 44.0  |
> |    MuSc($s=1$)    |   ViT-L-14-336    |    no    |        998.8        |   7168   |    97.8     | 93.8  |
> |    MuSc($s=2$)    |   ViT-L-14-336    |    no    |        605.8        |   5666   |    97.1     | 93.8  |
> |    MuSc($s=3$)    |   ViT-L-14-336    |    no    |        513.5        |   5026   |    96.7     | 93.7  |
> | APRIL-GAN(0-shot) | ViT-B-16-plus-240 |   yes    |        64.7         |   3226   |      -      |   -   |
> |  WinCLIP(0-shot)  | ViT-B-16-plus-240 |    no    |         389         |    -     |    91.8     | 64.6  |
> |    MuSc($s=1$)    | ViT-B-16-plus-240 |    no    |        455.3        |   3002   |    95.4     | 91.9  |
> |    MuSc($s=2$)    | ViT-B-16-plus-240 |    no    |        285.7        |   2740   |    95.4     | 91.5  |
> |    MuSc($s=3$)    | ViT-B-16-plus-240 |    no    |        238.3        |   2684   |    95.2     | 91.4  |

---

> > ### Author Response · Authors · 2023-11-22
> > **Response to Reviewer NLA8 (Part 2)**
> >
> > **Detailed statistics on time consumption and memory usage.** Beyond the above modifications, according to your suggestion, we have further researched the potential solutions for acceleration and memory reduction. Before that, we made statistics on the time consumption and memory usage of each module.
> >
> > The time consumption of our method can be divided into three components: the backbone forward, the LNAMD module, and the MSM module. The RsCIN module only uses 0.7ms per image so it is not discussed. We have analyzed the time consumption per image (ms) of each component, as shown in the table below. In the case of MuSc($s=1$) with the ViT-L-14-336 backbone, the backbone forward, LNAMD, and MSM takes 101.9ms, 106.5ms, and 790.4ms per image, respectively. The MSM module significantly contributes to the overall inference time. This is because the MSM performs mutual scoring on each combination of stage $l$ and aggregation degree $r$. The Aggregation degree ($r=1$), ($r=3$), and ($r=5$) of the four stages take 251.9ms, 261.7ms, and 276.8ms per image, respectively.
> >
> > |   Methods   | Backbone Network  | Backbone forward (ms) |       | LNAMD (ms) |       |       | MSM (ms) |       | Total (ms) |
> > | :---------: | :---------------: | :-------------------: | :---: | :--------: | :---: | :---: | :------: | :---: | :--------: |
> > |             |                   |                       | $r=1$ |   $r=3$    | $r=5$ | $r=1$ |  $r=3$   | $r=5$ |            |
> > | MuSc($s=1$) |   ViT-L-14-336    |         101.9         | 35.4  |    34.7    | 36.4  | 251.9 |  261.7   | 276.8 |   998.8    |
> > | MuSc($s=2$) |   ViT-L-14-336    |         107.0         | 44.6  |    37.9    | 42.3  | 122.6 |  124.7   | 126.7 |   605.8    |
> > | MuSc($s=3$) |   ViT-L-14-336    |         117.1         | 47.8  |    31.9    | 44.8  | 89.9  |   90.3   | 91.6  |   513.5    |
> > | MuSc($s=1$) | ViT-B-16-plus-240 |         59.5          |  8.4  |    8.0     |  7.2  | 123.6 |  124.7   | 123.9 |   455.3    |
> > | MuSc($s=2$) | ViT-B-16-plus-240 |         69.4          | 10.2  |    6.2     |  7.2  | 63.6  |   64.3   | 64.7  |   285.7    |
> > | MuSc($s=3$) | ViT-B-16-plus-240 |         78.6          | 11.6  |    7.1     |  7.8  | 43.8  |   44.3   | 45.1  |   238.3    |
> >
> > The memory usage of our method mainly comes from the backbone and the LNAMD module, as shown in the table below. LNAMD uses the features of unlabeled images extracted by ViT. These features are aggregated with different aggregation degrees $r\in \{1, 3, 5\}$, and saved for later use. The aggregated features in LNAMD consume a significant amount of memory. MSM costs a small amount of memory, while backbone consumes a lot of memory. From the data in the table, it is clear that the size of backbone and the number of unlabeled images are the two parameters heavily relevant to memory usage. When we replace the backbone ViT-L-14-336 with a smaller ViT-B-16-plus-240 in MuSc ($s=1$), the memory usage is reduced from 7168MB to 3002MB. In addition, using fewer unlabeled images reduces the memory cost in MuSc with the ViT-L-14-336 backbone from 7168MB ($s=1$) to 5026MB ($s=3$).
> >
> > |   Methods   | backbone Network  | Backbone (MB) | LNAMD (MB) | MSM (MB) | Total cost/ maximum GPU  cost(MB) |
> > | :---------: | :---------------: | :-----------: | :--------: | :------: | :-------------------------------: |
> > | MuSc($s$=1) |   ViT-L-14-336    |     2400      |    3504    |   454    |             6358/7168             |
> > | MuSc($s$=2) |   ViT-L-14-336    |     2400      |    2460    |   272    |             5132/5666             |
> > | MuSc($s$=3) |   ViT-L-14-336    |     2400      |    2112    |   202    |             4714/5026             |
> > | MuSc($s$=1) | ViT-B-16-plus-240 |     1592      |    1114    |   132    |             2832/3002             |
> > | MuSc($s$=2) | ViT-B-16-plus-240 |     1592      |    1014    |    60    |             2666/2740             |
> > | MuSc($s$=3) | ViT-B-16-plus-240 |     1592      |    966     |    40    |             2598/2648             |

---

> > > ### Author Response · Authors · 2023-11-22
> > > **Response to Reviewer NLA8 (Part 3)**
> > >
> > > **Potential solutions to speeding up and reducing memory.** According to the statistics above, the backbone size and the amount of test set are the main reasons of the long inference time and high memory cost. This is why we divide the test set into subsets, which is discussed in Tab. 6 and Appendix A.4. Note that, in the setting with the minimum time consumption (238.3ms) and GPU memory (2684MB), MuSc surpasses WinCLIP by +3.4% image-AUROC and +26.8% PRO.
> > >
> > > Beyond the originally presented solution, we explore two other potential solutions. Firstly, using a smaller size backbone can significantly decrease both inference time and memory cost at the expense of slightly reduced performance. We plan to further explore discriminative features to achieve high accuracy using lightweight networks. Secondly, we plan to combine patch correlations across images with patch correlations within images, so as to further compress the number of test sets while maintaining high accuracy. Thanks for your valuable comments!
> > >
> > > **Q2: The authors should provide more detailed theories for MSM and RsCIN.**
> > >
> > > A2: We have added a more detailed theoretical explanation of the MSM module in Appendix A.1.1, and we introduce the theory of RsCIN module in detail in Appendix A.1.2.

---

### Meta-Review · Area_Chair_NSp3 · 2023-12-03

**Metareview:**

The paper proposes a patch-based zero-shot anomaly classification and segmentation method for industrial images. The proposed method is based on the assumption that the test set contains both unlabeled anomaly and normal images and employs scoring mechanisms to classify and segment anomalies in the images.

Overall, three out of five reviewers provided a positive review, although not very strong, and they think that the proposed method is novel and achieves state-of-the-art (SOTA) results. The comments from these reviewers were addressed by the authors during the rebuttal process; however, the reviewers didn't acknowledge whether they are satisfied with the authors' response.

Two reviewers, Reviewer 52S5 and Reviewer dCfU, were not in favor of the paper. Both reviewers think that the algorithm is incremental with not much justification for why certain heuristics were employed in the proposed method. They also raise concerns regarding the running time of the method during the inference. The comments from these reviewers were addressed by the authors during the rebuttal process; however, the reviewers didn't acknowledge whether they are satisfied with the authors' response.

Given the above discussion and rebuttal/changes to the paper, I recommend acceptance. This is mainly due to the reason that the proposed method achieves SOTA results on an important real-world problem.

**Justification For Why Not Higher Score:**

Although the proposed method achieves SOTA results, it appears to be an incremental improvement with several heuristics.

**Justification For Why Not Lower Score:**

The idea is interesting and the proposed method achieves SOTA results on an important real-world problem.

---

### Decision · Program_Chairs · 2024-01-16

Accept (poster)